# Altered GC- and AT-biased genotypes of *Ophiocordyceps sinensis* in the stromal fertile portions and ascospores of natural *Cordyceps sinensis*

**Yu-Ling Li[1][☯], Ling Gao[2][☯], Yi-Sang Yao[2][☯], Xiu-Zhang Li[1], Zi-Mei Wu[3], Ning-Zhi Tan[2], Zhou-Qing Luo[3][¤], Wei-Dong Xie[2], Jian-Yong Wu[4,5], Jia-Shi Zhu[1,2]***

**1** State Key Laboratory of Plateau Ecology and Agriculture, Qinghai Academy of Animal and Veterinary Sciences, Qinghai University, Xining, Qinghai, China, **2** Shenzhen Key Laboratory of Health Science and Technology, Institute of Biopharmaceutical and Health Engineering, Shenzhen International Graduate School, Tsinghua University, Shenzhen, Guangdong, China, **3** School of Life Sciences, Tsinghua University, Beijing, China, **4** State Key Laboratory of Chinese Medicine and Molecular Pharmacology, Shenzhen, Guangdong, China, **5** Department of Applied Biology and Chemistry Technology, The Hong Kong Polytechnic University, Hung Hom, Hong Kong

☯ These authors contributed equally to this work.
¤ Current address: State Key Laboratory of Cellular Stress Biology, School of Life Sciences, Faculty of Medicine and Life Sciences, Xiamen University, Xiamen, China
* zhujosh@163.com

**Data Availability Statement:** All relevant data are within the paper and its Supplementary file. The sequence data were from the GenBank database

## Abstract

### Objective

To examine multiple genotypes of *Ophiocordyceps sinensis* in a semi-quantitative manner in the stromal fertile portion (SFP) densely covered with numerous ascocarps and ascospores of natural *Cordyceps sinensis* and to outline the dynamic alterations of the coexisting *O. sinensis* genotypes in different developmental phases.

### Methods

Mature *Cordyceps sinensis* specimens were harvested and continuously cultivated in our laboratory (altitude 2,254 m). The SFPs (with ascocarps) and fully and semi-ejected ascospores were collected for histological and molecular examinations. Biochip-based single nucleotide polymorphism (SNP) MALDI-TOF mass spectrometry (MS) was used to genotype multiple *O. sinensis* mutants in the SFPs and ascospores

### Results

Microscopic analysis revealed distinct morphologies of the SFPs (with ascocarps) before and after ascospore ejection and SFP of developmental failure, which, along with the fully and semi-ejected ascospores, were subjected to SNP MS genotyping analysis. Mass spectra showed the coexistence of GC- and AT-biased genotypes of *O. sinensis* that were genetically and phylogenetically distinct in the SFPs before and after ejection and of developmental failure and in fully and semi-ejected ascospores. The intensity ratios of MS

and we have provided hyperlinks to the database in Supplementary S2 Table.

**Funding:** This research was supported by a grant from the Science and Technology Department of Qinghai Province, China, grant number 2021-SF-A4 "Study on key technologies of conservation of natural resource and industrial upgrading of Cordyceps sinensis", the major science and technology projects in Qinghai Province. The funders had no role in study design, data collection and analysis, decision to publish, or preparation of the manuscript.

**Competing interests:** The authors have declared that no competing interests exist.

peaks were dynamically altered in the SFPs and the fully and semi-ejected ascospores. Mass spectra also showed transversion mutation alleles of unknown upstream and downstream sequences with altered intensities in the SFPs and ascospores. Genotype #5 of AT-biased Cluster-A maintained a high intensity in all SFPs and ascospores. An MS peak with a high intensity containing AT-biased Genotypes #6 and #15 in pre-ejection SFPs was significantly attenuated after ascospore ejection. The abundance of Genotypes #5–6 and #16 of AT-biased Cluster-A was differentially altered in the fully and semi-ejected ascospores that were collected from the same *Cordyceps sinensis* specimens.

## Conclusion

Multiple *O. sinensis* genotypes coexisted in different combinations with altered abundances in the SFPs prior to and after ejection, the SFP of developmental failure, and the two types of ascospores of *Cordyceps sinensis*, demonstrating their genomic independence. Metagenomic fungal members present in different combinations and with dynamic alterations play symbiotic roles in different compartments of natural *Cordyceps sinensis*.

## Introduction

Natural *Cordyceps sinensis* is one of the most highly valued therapeutic agents in traditional Chinese medicine (TCM), with a centuries-long history of clinical use as a folk medicine administered for "Yin-Yang" double invigoration, health maintenance, disease amelioration, and post-disease and post-surgery recovery [1, 2]. The therapeutic profiles of *Cordyceps sinensis* for respiratory, cardiovascular, endocrine, kidney, and liver systems, etc., have been studied using modern pharmacological approaches and summarized [2, 3]. The lifespan-extending properties of natural *Cordyceps sinensis* and its mycelial fermentation products were validated in mice in a 4-year pharmacology study, and the molecular mechanisms have been explored using a whole-gene expression biochip technique [4, 5].

The Chinese Pharmacopeia defines natural *Cordyceps sinensis* as an insect-fungal complex consisting of the *Ophiocordyceps sinensis* fruiting body and a dead Hepialidae moth larva, *i.e.*, natural *Cordyceps sinensis* is not equal to the *O. sinensis* fungus [6–11]. Anatomy studies have reported that the caterpillar body of *Cordyceps sinensis* contains an intact larval intestine, an intact and thick larval body wall with numerous bristles, head tissues, and fragments of other larval tissues [7, 10, 11]. Yao and Zhu [6] and Li et al. [7] addressed the controversy surrounding the indiscriminate use of Latin names for natural insect-fungal complexes and multiple anamorphic-teleomorphic fungi. In this context, we temporarily refer to the fungus/fungi as *O. sinensis* and continue the customary use of the name *Cordyceps sinensis* to refer to the wild or cultivated insect-fungal complex, although this practice will very likely be replaced by the discriminative use of exclusive Latin names for different biological entities in the future.

Natural *Cordyceps sinensis* grows only in alpine areas above 3,000–3,500 m on the Qinghai-Tibetan Plateau and has a complex lifecycle [9–11]. Its maturation stages greatly impact its compartmental mycobiota profile [12–18], competitive proliferation of *Hirsutella sinensis*-like fungus/fungi [14], metagenomic polymorphism and diversity [19–26], differential coexistence and dynamic alterations of multiple genotypes of *O. sinensis* [13, 19–23], metatranscriptomic and proteomic expression [27, 28], chemical constituent fingerprint [14], and therapeutic efficacy and potency of the caterpillar body and stroma of *Cordyceps sinensis* as a natural

therapeutic agent [29]. Mycologists have identified 22 species from 13 fungal genera in natural *Cordyceps sinensis* [30], and molecular methodologies have revealed >90 species from >37 fungal genera [15, 16] and 17 genotypes of *O. sinensis* [7, 9–11, 31] and demonstrated the predominance of different fungi and metagenomic fungal diversity in the stroma and caterpillar body of natural and cultivated *Cordyceps sinensis* [15, 16, 19–23, 31].

Wei et al. [32] hypothesized that *H. sinensis* is the sole anamorph of *O. sinensis* based on the collection of 3 lines of evidence: (1) frequent isolation and mycological identification according to sporulation, conidial morphology and growth characteristics [30]; (2) microcycle conidiation of the ascospores [33–35]; and (3) systematic molecular analyses based on internal transcribed spacer (ITS) sequencing and random molecular marker polymorphism assays [32, 36–39]. The sole *O. sinensis* anamorph hypothesis for *H. sinensis* [32] has been widely accepted because the 3 lines of evidence meet the first and second criteria of Koch's postulates. Wei et al. [40] published a successful industrial artificial cultivation project reportedly using 3 GC-biased Genotype #1 *H. sinensis* strains as inoculants, seemingly satisfying the third criterion of Koch's postulates. However, the authors reported the detection of a single teleomorphic *O. sinensis* fungus in the fruiting body of cultivated *Cordyceps sinensis*, the sequence of which was reported to be identical to AT-biased Genotype #4 AB067749. Many studies have proven that the sequences of all AT-biased *O. sinensis* genotypes are not present in the 5 genome assemblies of GC-biased Genotype #1 *H. sinensis* Strains 1229, CC1406-203, Co18, IOZ07, and ZJB12195 [41–45] but instead belong to the genomes of independent *O. sinensis* fungi [7–11, 13, 31]. Thus, the sequence mismatching between the inoculants and teleomorphic *O. sinensis* in cultivated *Cordyceps sinensis* products reported by Wei et al. [40] disproves the sole anamorph hypothesis for *H. sinensis* according to the fourth criterion of Koch's postulates.

Teleomorphs of *O. sinensis* found in natural *Cordyceps sinensis* specimens collected from different geographic areas have been reported to belong to Genotypes #1 [37, 38, 40], #3 [46], #4 [40], either #4 or #5 [37, 47] and #7 [48] without coexisting with other genotypes of *O. sinensis*. Wei et al. [40] reported teleomorphs of GC-biased Genotype #1 and AT-biased Genotype #4 of *O. sinensis* in natural and cultivated *Cordyceps sinensis*, respectively. Li et al. [49] reported the detection of both GC-biased Genotype #1 and AT-biased Genotype #5 in 25-day cultures of teleomorphic monoascospores of natural *Cordyceps sinensis*. Bushley et al. [50] reported multicellular heterokaryotic mono-, bi-, and tri-nucleated structures of *Cordyceps sinensis* hyphae and ascospores. Li et al. [31] reported the differential coexistence of multiple GC- and AT-biased genotypes of *O. sinensis* in the immature and mature stromata, stromal fertile portion (SFP) that was densely covered with numerous ascocarps, and ascospores of natural *Cordyceps sinensis*, which were altered during *Cordyceps sinensis* maturation. These findings regarding teleomorphic *O. sinensis* indicate that the independent metagenomic members of natural *Cordyceps sinensis* alternate dynamically and play symbiotic roles in different developmental and maturational stages of the *Cordyceps sinensis* lifecycle, constituting a natural integrated microecosystem [29].

In this study, we examined the microscopic morphology of SFPs prior to and after ascospore ejection and SFP of developmental failure. We also monitored the dynamic alterations of *O. sinensis* genotypes in a semiquantitative manner in the SFP samples and in the fully and semi-ejected ascospores using the biochip-based MassARRAY single nucleotide polymorphism (SNP) MALDI-TOF mass spectrometry (MS) genotyping technique. We observed differential co-occurrence and dynamic alterations of *O. sinensis* genotypes in the SFPs and ascospores of *Cordyceps sinensis*.

## Materials and methods

### Collection of *Cordyceps sinensis* specimens

*Cordyceps sinensis* specimens were purchased in local markets from the Hualong (located at 36˚13'N, 102˚19'E; altitude 3553.1 m) and Yushu areas (located at 33˚01'N, 96˚48'E; altitude 4457.6 m) of Qinghai Province, China. Governmental permission was not required for *Cordyceps sinensis* purchases in local markets, and the collections of *Cordyceps sinensis* specimens from sales by local farmers fall under the governmental regulations for traditional Chinese herbal products.

Mature *Cordyceps sinensis* specimens were collected in mid-June. These specimens showed a plump caterpillar body and a long stroma (>5.0 cm) with the formation of an expanded fertile portion close to the stromal tip (*cf*. Fig 3.1 below), which was densely covered with ascocarps (*cf*. Fig 3.2) [25, 31].

Some specimens used in the histology examination were washed thoroughly on site in running water with gentle brushing, soaked in 0.1% mercuric chloride for 10 min for surface sterilization and washed 3 times with sterile water. The thoroughly cleaned specimens were immediately frozen in liquid nitrogen on site and kept frozen during transportation to the laboratory and during storage prior to further processing [14, 19, 31].

### Continued cultivation of mature *Cordyceps sinensis* specimens

Some mature *Cordyceps sinensis* specimens were harvested along with the outer mycelial cortices and soil surrounding the caterpillar body, replanted in paper cups in soil obtained from *Cordyceps sinensis* production areas and cultivated in our laboratory (altitude 2,254 m; longitude and latitude of 101˚45' E 36˚40' N) in Xining City, Qinghai Province of China (*cf*. Fig 1 of [31] and S5 Fig). Because of the phototropism of natural *Cordyceps sinensis*, we kept the windows fully open, allowing sufficient sunshine and a natural plateau breeze blowing over the cultivated specimens in the paper cups. The room temperature was maintained naturally, fluctuating with the lowest temperature at 18–19˚C during the night and the highest temperature at 22–23˚C in the early afternoon. The relative humidity of our laboratory was maintained at approximately 30%–40%, and the absolute humidity on the surface of *Cordyceps sinensis* stromata was maintained at up to 90%–100% by spraying water using an atomizer twice a day in the morning and evening [31].

### Collection and preparation of *Cordyceps sinensis* ascospores

Fig 1 of [31] (S5 Fig) shows a collection of the fully ejected ascospores of *Cordyceps sinensis* using double layers of autoclaved weighing paper. It also showed numerous ascospores adhered to the outer surface of asci during massive ascospore ejection, which failed to be brushed off for collection using an autoclaved brush; hence, these ascospores were instead gently scratched off using a disinfected inoculation shovel or ring and referred to as semi-ejected ascospores.

The fully and semi-ejected ascospores were cleaned separately by 2 washes with 10% and 20% sucrose solutions and 10-min centrifugation at 1,000 rpm (desktop centrifuge, Eppendorf, Germany); the supernatant was discarded after each centrifugation. The pellets (ascospores) were subsequently washed with 50% sucrose solution and centrifuged for 30 min, and the ascospores that floated on the liquid were collected [31, 51]. The 2 types of ascospores were stored in a -80˚C freezer prior to further processing.

## Collection of *Cordyceps sinensis* SFPs

The SFPs (*cf.* Fig 3.1 below) were sampled prior to and after ascospore ejection. The SFP of developmental failure that showed no sign of ascospore ejection during prolonged cultivation were also sampled. These SFP samples were stored in an -80˚C freezer prior to further processing for DNA extraction.

## Reagents

Laboratory common reagents, such as ethanol, sucrose, paraformaldehyde, hematoxylin, eosin, agarose and electrophoresis reagents, were purchased from Beijing Bioland Technology Company. The Universal DNA Purification kit was a product of the TIANGEN Biotech Company, China. The DNeasy Plant Mini Kit was a product of Qiagen Company, Germany. The *Taq* PCR reagent kit and Vector NTI Advance 9 software were purchased from Invitrogen, United States.

Reagents and biochips required for MassARRAY SNP MALDI-TOF MS genotyping were from Sequenom, USA (acquired by Agena Bioscience Company). The SNP MALDI-TOF MS genotyping experiments were performed at CapitalBio Co., Beijing, using a 384-well biochip (SpectroCHIP, Sequenom) and a commercial "iLEX Gold Cocktail" (Sequenom; exact formula unavailable) containing all necessary PCR reagents, including polymerase and dNTPs, and dideoxy A, T, C, and G terminator nucleotides (ddNTPs) to control DNA chain elongation.

## Histological examination of the SFP, ascocarps and ascospores of *Cordyceps sinensis*

Some of the mature *Cordyceps sinensis* stromal specimens collected prior to and after the massive ejection of ascospores and of developmental failure were immersed in 10% formalin for fixation and subjected to dehydration in 50%, 70% and 95% ethanol for 1 h each [52]. The SFP tissues (with numerous ascocarps) were embedded in paraffin and sliced to 5-μm thickness (Model TN7000 Microtome, Tanner Scientific, Germany) [31, 52]. The ascus slices were stained with hematoxylin-eosin (HE) and observed under optical and confocal microscopes (Model Primo Star and Model LSM 780, ZEISS, Germany).

## Sample preparation and extraction of genomic DNA

The frozen SFP samples were individually ground to a powder in liquid nitrogen, including the SFPs (with ascocarps) (1) prior to ascospore ejection, (2) after ascospore ejection, and (3) of developmental failure, each of which was a combination of 5 specimens. The ascospores (4) fully ejected and (5) semi-ejected were collected from 5 *Cordyceps sinensis* specimens and treated in a lysate solution before extracting genomic DNA [31]. Genomic DNA was individually extracted from these SFP powders and lysed ascospores using the DNeasy Plant Mini Kit (Qiagen, Germany) according to the manufacturer's manual [14, 25, 31] and quantified by measuring the absorbance at $UV_{260}$ (Eppendorf BioSpectrometer®; Hamburg, Germany).

## Primer design for the first-step PCR in SNP genotyping analysis

The primers *Hsprp1*, *Hsprp3*, *ITS4*, *P1*, *P2* and *P4*, which were designed, verified, used, and reported previously [14, 19–23], were used in the first-step PCR (Table 1 and Fig 1) to amplify the ITS1-5.8S-ITS2 segments according to the following protocol: (1) 95˚C for 5 min; (2) 35 cycles of 95˚C for 30 s, 55˚C for 30 s, and 72˚C for 1 min; (3) 72˚C for 10 min; with final holding at 4˚C [14, 19]. The primer pairs *Hsprp1/3* and *Hsprp1/ITS4* were favorable for amplifying the ITS sequences of GC-biased genotypes, and the primer pairs *P1/P2*, *P2/P1* and *P2/P4* were

designed to amplify the sequences of AT-biased genotypes (the GC and AT content information of the 17 *O. sinensis* genotypes is listed in S1 Table). All primers were synthesized by Invitrogen Beijing Lab. or Beijing Bomaide Technology Co. Amplicons from the first-step PCR were quantified by measuring the absorbance at $UV_{260}$ and examined by agarose gel electrophoresis.

## Phylogenetic analysis of multiple genotypes of *O. sinensis*

GenBank accession numbers of 17 genotypes of *O. sinensis* are listed in S2 Table with hyperlinks to the GenBank database and were analyzed by conferring a phylogenetic tree using the maximum likelihood (ML) algorithm with MEGA software (Fig 2) [53] to assist in the design and analysis of the SNP genotyping experiments. The ITS1-5.8S-ITS2 segment sequence ranges in 5 genomes of Genotype #1 *H. sinensis* Strains 1229, CC1406-203, Co18, IOZ07, and ZJB12195 [41–45], either a single ITS copy or multiple repetitive ITS copies (S2 Table), were entered into the ML phylogenetic analysis.

## Extension primer design for MassARRAY SNP MS genotyping analysis based on the phylogenetic relationships of multiple genotypes of *O. sinensis*

In addition to the GC-biased genotype clade (side-noted in blue in Fig 2), the AT-biased genotype clade (side-noted in red) contains 2 AT-biased branches [7–11, 31]: AT-biased Cluster-A consists of Genotypes #5–6 and #16–17, and AT-biased Cluster-B consists of Genotypes #4 and #15. The representative sequences of GC-biased Genotype #1 AB067721, Genotype #5 AB067740 within AT-biased Cluster-A, and Genotype #4 AB067744 within AT-biased Cluster-B were used as the reference sequences for designing SNP extension primers (Table 2) for biochip-based MassARRAY SNP MALDI-TOF MS genotyping. Extension primers 067721–477 and 067721–531 were used to distinguish between the GC- and AT-biased genotypes but not between the multiple AT-biased genotypes (*cf.* Fig 1) [19, 21, 22]. Extension primers 067740–324, 067744–324 (reverse complement), and 067740–328 (reverse complement) were used to distinguish between the multiple AT-biased genotypes using a stratification genotyping strategy.

## Biochip-based MassARRAY SNP MALDI-TOF MS genotyping

ITS sequences were amplified using the aforementioned primer pairs from genomic DNA prepared from SFPs prior to and after ejection and of developmental failure and fully and semi-ejected ascospores. The amplicons obtained from the first-step PCR were subjected to neutralization of the unincorporated dNTPs with shrimp alkaline phosphatase and used as templates for single nucleotide extension toward the selected SNP sites shown in Fig 1. These amplicons were mixed with an iLEX Gold cocktail (Sequenom) containing one of the extension primers listed in Table 2 per reaction and amplified according to the following protocol: (1) 94°C for 30 s; (2) 94°C for 5 s; (3) 52°C for 5 s; (4) 80°C for 5 s; (5) return to (3) for 4 more times (for a total of 5 "smaller" cycles); (6) return to (2) for 39 more times (for a total of 40 "larger" cycles); (7) 72°C for 3 min; and (8) 4°C indefinitely [19, 21, 22]. After purification, the extended products with one of the mass-modified nucleotides were analyzed using a 384-well SpectroCHIP (Sequenom), and their masses were examined using a MassARRAY analyzer to determine heterogeneous genotypes [19, 21, 22, 54]. Each primer-extension MS genotyping test for the combined 5 samples was repeated at least twice, and the genotyping results were analyzed using software TYPER 4.0 (Sequenom). The MassARRAY SNP MALDI-TOF MS genotyping experiments were performed by a service company, CapitalBio Co., in Beijing, China.

**Table 1. "Universal" primer and genotype-specific primers for the first-step PCR amplification of ITS segments.**

| Primer Name | Direction | Primer sequence |
| --- | --- | --- |
| For amplification of the ITS sequences of GC-biased Genotype #1 | | |
| *Hsprp1* | Forward | ATTATCGAGTCACCACTCCCAAACCCCC |
| *Hsprp3* | Reverse | CGAGGTTCTCAGCGAGCTACT |
| *ITS4* | Reverse | TCCTCCGCTTATTGATATGC |
| For amplification of the ITS sequences of AT-biased genotypes | | |
| *P1* (AB067740$_{260-279}$, AB067744$_{256-275}$) | Forward | ACGCAGCGAAATACGATAAG |
| *P2* (AB067740$_{356-338}$, AB067744$_{352-334}$) | Reverse | CATGCCCGCTAGAGTGCTA |
| *P4* (AB067740$_{260-279}$, AB067744$_{256-267}$) | Forward | ACGCAGCGAAATGCAATAAG |

## Results

### Microscopy observations of histological structures of the SFPs

Fig 3.1 shows a mature *Cordyceps sinensis* specimen labeled for the SFP (the stromal fertile portion), and Fig 3.2 shows a confocal image (bar: 1 mm) of a transverse section of the SFP of *Cordyceps sinensis*, which is densely covered by numerous ascocarps (with a red circle in Fig 3.2 indicating one of the multiple ascocarps). Fig 3.3 shows an optical microscopy image (40x) of an HE-stained *Cordyceps sinensis* ascocarp, which contains multiple ascospores prior to ejection. Fig 3.4 shows a confocal image (bar: 100 μm) of 2 ascocarps; the upper ascocarp is a maturing ascocarp with not yet opened perithecium containing multiple ascospores prior to ejection, and the lower ascocarp is a fully mature ascocarp during ascospore ejection with multiple ascospores aggregating toward the opening of the perithecium. Fig 3.5 shows a confocal image close to the opening of the perithecium during ascospore ejection with an arrow pointing to a semi-ejected ascospore hanging out of the opening of the perithecium (bar: 20 μm). Fig 3.6 shows an optical microscopy image (10x) of HE-stained SFP after ascospore ejection. Fig 3.7 (10x) and 3.8 (40x) show optical images of HE-stained SFPs of developmental failure with abnormal structures of ascocarps (compared to normal structures shown in Fig 3.3–3.4) and containing no normally developed ascospores.

In addition, Fig 1 of [31] (reproduced and shown in S5 Fig) shows the numerous semi-ejected ascospores that tightly adhered to the outer surface of asci. Fig 3 of [31] shows the microscopic images of fully ejected *Cordyceps sinensis* ascospores with or without calmodulin fluorescent staining.

### Phylogenetic analysis of 17 genotypes of *O. sinensis* and multiple repetitive ITS copies in the genome of Genotype #1 *H. sinensis*

The ML phylogenetic tree (*cf.* Fig 2) shows the phylogeny of the 17 GC- and AT-biased genotypes of *O. sinensis* (the GC and AT contents of the genotypes are shown in S1 Table) and the genome ITS sequences of Genotype #1 *H. sinensis*, including multiple repetitive ITS copies in the genome assemblies JAAVMX000000000 and NGJJ00000000 and a single copy each in ANOV00000000, LKHE00000000 and LWBQ00000000 of Genotype #1 *H. sinensis* Strains IOZ07, CC1406-203, Co18, 1229 and ZJB12195, respectively [41–45]. The topology of the ML tree was verified with Bayesian majority rule consensus trees [7, 11, 31, 55], which were inferred using MrBayes v3.2.7 software (the Markov chain Monte Carlo [MCMC] algorithm) [56]. Notably, each of the genome assemblies ANOV00000000, LKHE00000000 and LWBQ00000000 of *H. sinensis* Strains Co18, 1229 and ZJB12195, respectively [41–43], contains only one ITS copy, indicating that other repetitive ITS copies might have been discarded

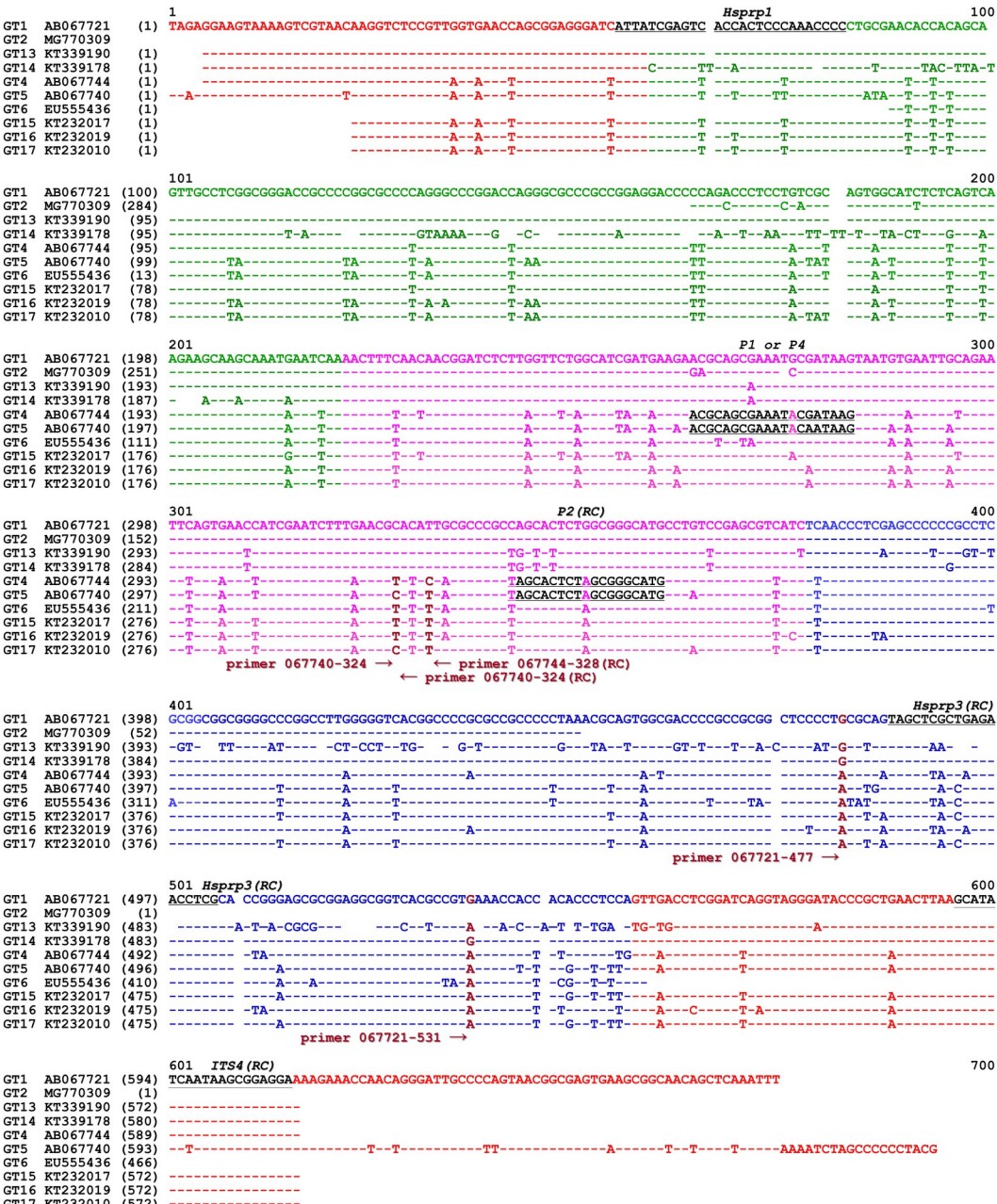

**Fig 1. Alignment of ITS sequences of GC- and AT-biased genotypes of *O. sinensis* and primer locations.** GT refers to genotype. Genotypes #1–2 and #13–14 are GC-biased *O. sinensis* genotypes, and Genotypes #4–6 and #15–17 are AT-biased *O. sinensis* genotypes. The GC and AT contents of the 17 *O. sinensis* genotypes are shown in S1 Table. The underlined sequences in black represent primers for the first-step PCR. Extension primers 067721–477, 067721–531, 067740–324, 067744–324 (RC, reverse complement) and 067740–328 (RC) in brown were used for SNP mass spectrometry genotyping to distinguish between genotypes of *O. sinensis*. Arrows indicate the directions of the primer extension reactions toward the SNP alleles in brown. "-" refers to an identical base, and spaces refer to unmatched sequence gaps.

during assembly of genome shotgun sequences. All these genomic ITS sequences are GC biases and were clustered in the GC clade along with all GC-biased genotypes labeled in blue alongside Fig 2, but very distant from the AT-biased clade labeled in red alongside. The genome repetitive ITS copies contain more or less multiple scattered insertion/deletion and

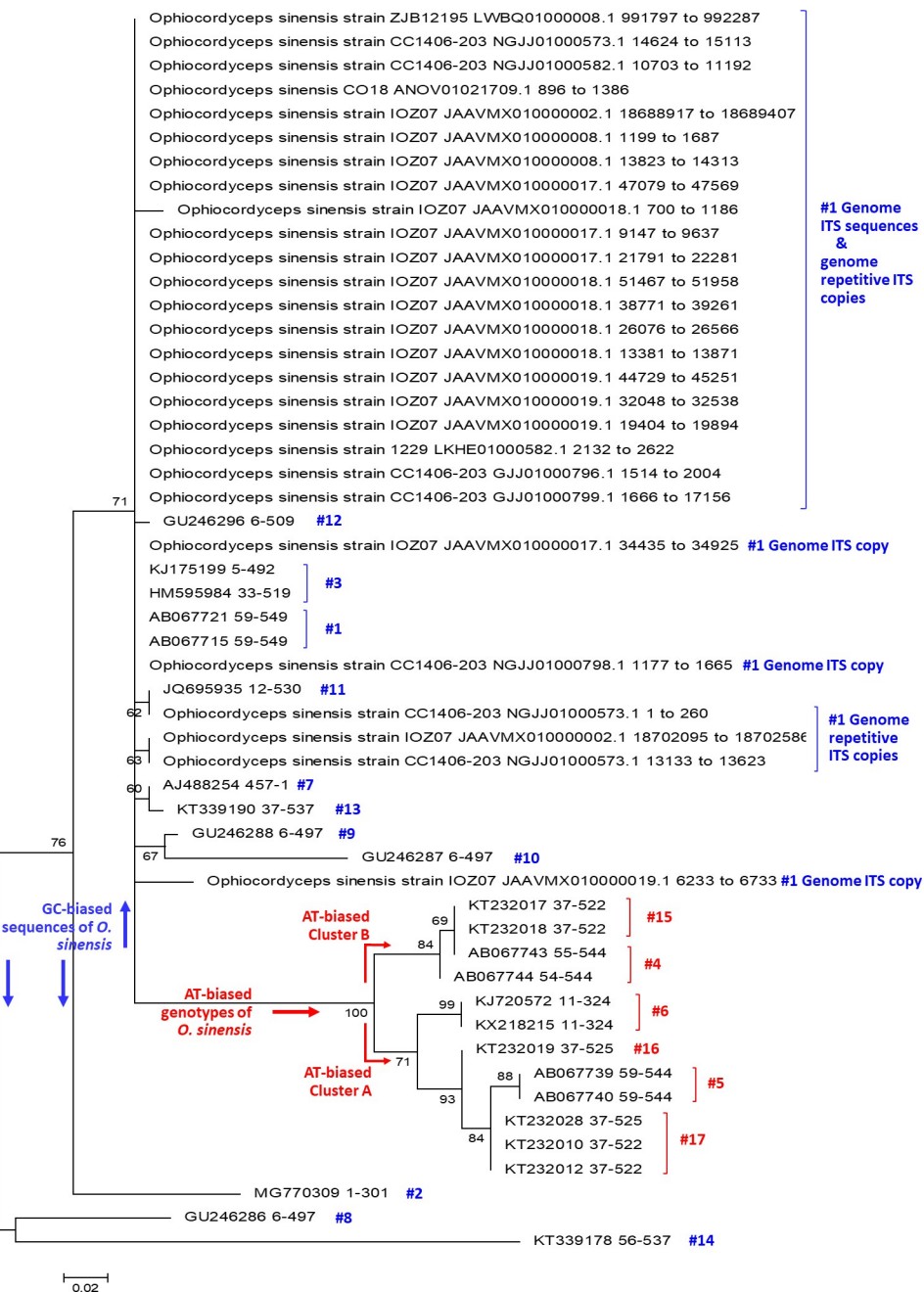

**Fig 2. A maximum likelihood (ML) phylogenetic tree.** The ML tree was inferred using MEGA software [53], containing 25 ITS sequences (representing 17 genotypes of *O. sinensis*) and 27 genomic ITS sequences (including multiple repetitive ITS copies) of Genotype #1 *H. sinensis* Strain 1229, CC1406-203, Co18, IOZ07, and ZJB12195. S2 Table lists the GenBank accession numbers that are hyperlinked to GenBank database, strain or isolate information and ITS1-5.8S-ITS2 sequence range for the 52 ITS sequences that were analyzed phylogenetically in Fig 2. The phylogenetic topology of the ML tree was verified according to the topologies of Bayesian majority-rule consensus phylogenetic trees [7,10, 11, 31].

transversion point mutations, but no or only a few transition point mutations [57]. The genetic and phylogenetic findings indicate that the genomic repetitive ITS copies were not the mutagenic targets of Repeat-Induced Point mutation (RIP) causing cytosine-to-thymine transition.

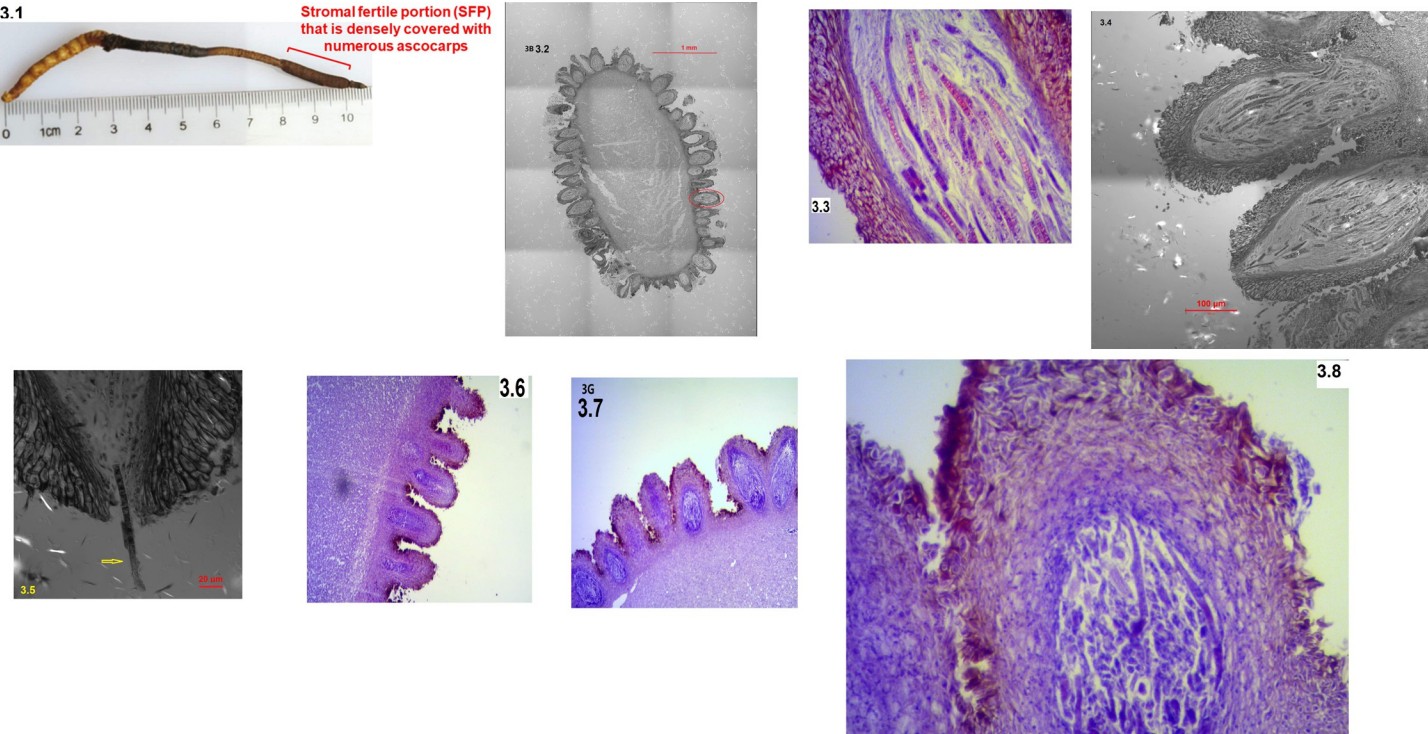

**Fig 3. Microscopic images of the SFPs (with ascocarps) of natural *Cordyceps sinensis*.** Panel 3.1 shows a photo of a mature *Cordyceps sinensis* specimen labeled with stromal fertile portion (SFP). Panel 3.2 shows a confocal image (bar: 1 mm) of a transverse section of the *Cordyceps sinensis* SFP densely covered with numerous ascocarps prior to ascospore ejection, with a red circle indicating one of the multiple ascocarps densely covering the SFP. Panel 3.3 shows an optical close-up image (40x) of an HE-stained ascocarp containing multiple ascospores of multicellular ascosporic cells. Panel 3.4 shows a confocal image (bar: 100 µm) of 2 ascocarps; the upper one is a maturing ascocarp, and the lower one is a mature ascocarp containing multiple ascospores aggregating toward the opening of the perithecium. Panel 3.5 shows a confocal image close to the opening of the perithecium and a semi-ejected ascospore hanging out of the opening of the perithecium (bar: 20 µm). Panel 3.6 shows an optical image (10x) of HE-stained SFPs after ascospore ejection. Panels 3.7 (10x) and 3.8 (40x) show optical images of HE-stained SFPs of developmental failure. The confocal images shown in Panels 3.2 and 3.4 are large images created by stitching together tile images automatically by the computer and software of the ZEISS LSM 780 confocal microscope, which allow extremely high resolution from designated large fields of view, because of limited scanning areas of the ZEISS LSM 780 confocal microscope.

All AT-biased genotypes feature multiple scattered transition point mutations (*cf*. Fig 1), among which Genotypes #5–6 and #16–17 were clustered into AT-biased Cluster-A and Genotypes #4 and #15 in AT-biased Cluster-B (*cf*. Fig 2). The genome assemblies of GC-biased Genotype #1 *H. sinensis* strains do not contain any AT-biased genotype sequences, indicating genomic independence of the *O. sinensis* genotypes.

## SNP genotyping to distinguish between GC- and AT-biased *O. sinensis* genotypes in SFPs before and after ascospore ejection and of developmental failure

Li et al. [31] reported the co-occurrence of GC-biased Genotype #1 *H. sinensis* and AT-biased Genotypes #4–6 and #15 of *O. sinensis* in SFPs based on the application of several pairs of genotype-specific primers and cloning-based amplicon sequencing approaches. The biochip-based MassARRAY SNP MALDI-TOF MS genotyping technique was used to examine the dynamic alterations of the *O. sinensis* genotypes in SFPs prior to and after ascospore ejection and of developmental failure in a semiquantitative manner. Equal amounts of genomic DNA (calculated based on the absorbance at $UV_{260}$) were applied to the first-step PCR using the primer pairs *Hsprp1/3* and *Hsprp1/ITS4* to amplify the ITS sequences of GC-biased genotypes

**Table 2. SNP extension primers for MassARRAY MALDI-TOF MS genotyping.**

| Primer Name | Primer sequence |
|---|---|
| Used in the second-step PCR designed based on the AB067721 sequence to amplify the GC-biased sequences of *O. sinensis* genotypes | |
| *067721–477* | CGCCGCGGCTCCCCT |
| *067721–531* | AGGCGGTCACGCCGT |
| Used in the second-step PCR designed based on both AB067744 and AB067740 sequences to amplify the AT-biased sequences of *O. sinensis* genotypes | |
| *067740–324* | GTAAACTATCGAATCTTTAAACG |
| *067744–324* (reverse complement) | GCTAGCGGGCGTAGTAT |
| *067740–328* (reverse complement) | GTGCTAGCGGGCGTA |

in greater quantities than those of AT-biased genotypes, and the primer pairs *P1/P2*, *P2/P1* and *P2/P4* were used to amplify the sequences of AT-biased genotypes (*cf.* Table 1 and Fig 1).

Extension primers 067721–477 and 067721–531 (*cf.* Table 2) were used in the second-step PCR, and the reactions extended the extension primers with a single nucleotide toward the SNPs at positions 477 and 531 in the AB067721 sequence (*cf.* Fig 1). Mass spectra showed 4 allelic peaks, Peaks A, C, G, and T, representing the extension reactions of the primers 067721–477 and 067721–531 with a single extended nucleotide adenine, cytosine, guanine, and thymine, respectively.

**SNP genotyping using extension primer 067721–477.** Fig 4.1 shows a mass spectrum of extension primer 067721–447 (*cf.* Table 2) for the SFP prior to ascospore ejection. Peak G with the highest MS intensity represents the GC-biased Genotype #1 *H. sinensis*. Peaks C, A, and T possess much lower intensities. Peak A represents AT-biased genotypes with transition point mutations (*cf.* Fig 1), and Peaks C and T represent 2 transversion point mutation genotypes of unknown upstream and downstream sequences of the allelic peaks. The primers employed for the first-step PCR preferentially amplified GC-biased genotype sequences, resulting in a greater quantity of GC-biased amplicon copies that served as the templates for the second-step PCR. Table 3 shows the intensity ratios of the MS peaks: G:A, G:C and G:T.

Fig 4.2 shows a mass spectrum for the postejection SFP. The intensities of Peaks A and G appeared to increase proportionally compared with the corresponding peaks shown in Fig 4.1, resulting in a similar G:A ratio (Table 3). In contrast, Peaks C and T were greatly attenuated, resulting in significant elevations in the intensity ratios of G:C and G:T (Table 3). The ejection-related increase in the GC- and AT-biased genotypes of *O. sinensis*, coupled with declines in the 2 unknown transversion mutation genotypes, indicate that these genotypes play different roles in SFPs prior to, during and after ascospore ejection.

Fig 4.3 shows the overlapping mass spectra of Fig 4.1 and 4.2 after adjusting and aligning the vertical axis for intensity and visualizes the dynamic alteration of the allelic peaks in the SFPs from pre-ejection to post-ejection.

S1 Fig shows a mass spectrum for an SFP of developmental failure. Compared with Peaks G and A shown in Fig 4.1, the intensity of Peak G in S1 Fig was elevated, while Peak A remained unchanged, resulting in a small elevation of the G:A intensity ratio (Table 3). Peak C was absent, while Peak T of the transversion point mutation genotype was lower than that for the pre-ejection SFP shown in Fig 4.1, resulting in a significant elevation of the G:T ratio (Table 3).

**SNP genotyping using extension primer 067721–531.** S2.1 Fig shows a mass spectrum of extension primer 067721–531 (Table 2) for the pre-ejection SFP, displaying allelic Peaks A and G (representing AT-biased genotypes of *O. sinensis* and GC-biased Genotype #1, respectively) with comparable intensities and resulting in a G:A intensity ratio of 1.06 (S3 Table). Because the primers employed for the first-step PCR preferentially amplified the sequences of

GC-biased genotypes, the comparable intensities of Peaks A and G might indicate a greater quantity of natural abundance of AT-biased genotypes than GC-biased Genotype #1 *H. sinensis* in the genomic DNA pool extracted from the pre-ejection SFP. While Peak C was absent, Peak T showed a very low intensity, representing a transversion mutation genotype of unknown upstream and downstream sequences and resulting in a high G:T ratio of 12.7 (S3 Table).

S2.2 and S2.3 Fig show similar topologies of the mass spectra for the SFPs after ascospore ejection and of developmental failure, respectively, displaying an elevated Peak G and a considerably attenuated Peak A relative to those shown in S2.1 Fig. These results indicated significant decreases in the amount of AT-biased genotypes of *O. sinensis* in the SFPs after ascospore ejection and of developmental failure, while GC-biased Genotype #1 *H. sinensis* was relatively increased. The dynamic bi-directional changes resulted in increases in the G:A intensity ratios to 19.3 and 13.5 for the SFPs after ejection and of developmental failure, respectively (S3 Table), compared with a G:A ratio of 1.06 obtained for the pre-ejection SFP. Peaks C and T are absent (S2.2 and S2.3 Fig).

S2.4 Fig shows the overlapping mass spectra of S2.1 and S2.2 Fig after adjusting and aligning the vertical axis for intensity and visualizes the dynamic alteration of the allelic peaks in the SFPs from pre-ejection to post-ejection.

## Stratified SNP genotyping to distinguish between the AT-biased genotypes of *O. sinensis* in the SFPs prior to and after ascospore ejection and of developmental failure

Li et al. [31] reported the detection of Genotypes #5–6 of AT-biased Cluster-A and Genotypes #4 and #15 of AT-biased Cluster-B in the SFP (*cf*. Fig 2). According to the sequence alignment shown in Fig 1, SNP MS genotyping was performed *via* a stratification strategy using extension primers 067740–324, 067744–324 (reverse complement) and 067740–328 (reverse complement) (*cf*. Table 2 and Fig 1) to distinguish between Genotypes #4–6 and #15. The templates of the extension reactions were the amplicons obtained from the first-step PCR using primer pairs *P1/P2*, *P2/P1* and *P2/P4* (*cf*. Table 1), which amplify the AT-biased sequences of *O. sinensis*.

**Using extension primer 067740–328 to distinguish between AT-biased Genotype #4 and other AT-biased genotypes.** S3.1 Fig shows a mass spectrum of extension primer 067740–328 (reverse complement; *cf*. Fig 1 and Table 2) for the SFP prior to ascospore ejection. The major Peak T represents the sum of the mass intensities of Genotypes #5–6 and #15. Although the extension reaction using primer 067740–328 could also produce an extended thymine on the DNA templates of Genotypes #16–17 (*cf*. Fig 1), these genotypes were absent in the SFPs of natural *Cordyceps sinensis* according to the results obtained using genotype-specific primers and PCR amplicon-cloning techniques [31]. Peak C was buried in a descending MS loop, representing a very low level of Genotype #4 (*cf*. Fig 1), resulting in a T:C intensity ratio of 14.1 (S4 Table). Peak G was nearly flat and negligible. Peak A with a low intensity represented a co-occurring transversion mutation genotype of unknown upstream and downstream sequences, resulting in a T:A ratio of 11.1 (S4 Table).

S3.2 and S3.3 Fig show mass spectra for the SFPs after ascospore ejection and of developmental failure with similar MS topologies to that shown in S3.1 Fig. Compared with S3.1 Fig, Peak T (Genotypes #5–6 and #15) remained the major allele peak with increased intensities in both S3.2 and S3.3 Fig. Peak C (Genotype #4) was still buried in the descending MS loop, resulting in a similar (S3.2 Fig) or an increased (S3.3 Fig) T:C ratio (S4 Table). Peak A (a transversion point mutation genotype of unknown upstream and downstream sequences) was

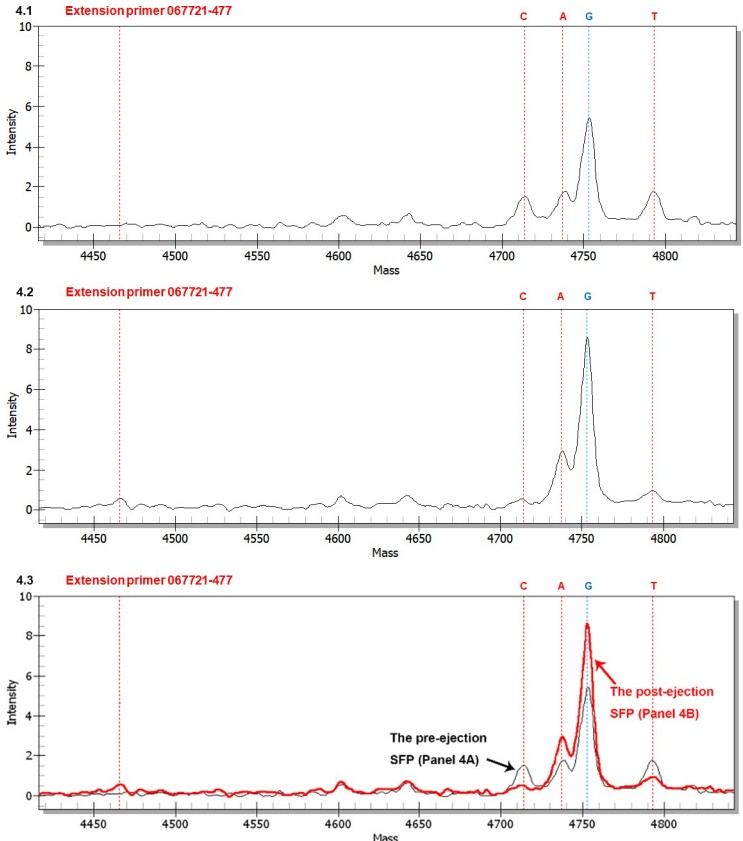

**Fig 4. MALDI-TOF mass spectra (MS) of extension primer 067721–477 to distinguish between GC- and AT-genotypes in the SFPs (with ascocarps) prior to and after ascospore ejection.** The extension reaction proceeded from extension primer 067721–477 toward the SNP at position 477 in the AB067721 sequence (*cf.* Fig 1 for the location). In the allelic peaks, "C" denotes extension of the primer with an extended cytosine; "A" indicates an extended adenine, "G" represents an extended guanine, and "T" refers to an extended thymine. Panels 4.1 and 4.2 show the mass spectra for the SFPs prior to and after ascospore ejection, respectively. Panel 4.3 shows the overlapping mass spectra for Panels 4.1 (black tracing) and 4.2 (red tracing) with alignment of the horizontal and vertical axes of the panels.

attenuated in S3.2 and S3.3 Fig relative to that shown in S3.1 Fig, resulting in increases in the T:A ratio (S4 Table).

**Table 3. Mass intensity ratios of the SNP peaks of transition and transversion mutation genotypes in the SFPs prior to and after ascospore ejection and of developmental failure.**

| Extension primer | Allelic ratio | Intensity ratio | | |
|---|---|---|---|---|
| | | **Prior-ejection SFP (*cf.* Fig 4.1)** | **Post-ejection SFP (*cf.* Fig 4.2)** | **SFP of developmental failure (*cf.* S1 Fig)** |
| 067721–477 | G:A | **3.3** (5.3÷1.6) | **3.0** (↔; 8.4÷2.8) | **3.9** (↑; 7.0÷1.8) |
| | G:C | **4.4** (5.3÷1.2) | **21.0** (↑↑↑↑; 8.4÷0.4) | — |
| | G:T | **3.3** (5.3÷1.6) | **9.3** (↑↑; 8.4÷0.9) | **10.0** (↑↑↑; 7.0÷0.7) |

Note: Peak G in Figs 4 and S1 represents GC-biased Genotype #1 *H. sinensis*; Peak A indicates AT-biased genotypes of *O. sinensis* (*cf.* Fig 1). Peaks C and T denote 2 transversion mutation genotypes of unknown upstream and downstream sequences. "↔" indicates no significant change (within 20% variation), and "↑" (increase less than two-fold), "↑↑" (increase greater than two-fold but less than three-fold), "↑↑↑" (increase greater than three-fold but less than four-fold) and "↑↑↑↑" (increase greater than four-fold) denote significant increases in intensity ratios compared to that in the pre-ejection SFP. "−" means that one of the allelic peaks was missing, and no ratio could be calculated.

Peak C with extremely low intensities shown in S3.1–S3.3 Fig for all 3 SFP samples indicated a consistently low quantity of Genotype #4 of AT-biased Cluster-B, consistent with the report by Li et al. [31] based on the use of genotype-specific primers and cloning-based PCR amplicon sequencing techniques.

**Using extension primer 067744–324 to distinguish between AT-biased Genotypes #5 and #15.**  After confirming consistently extremely low intensities of Peak C (Genotype #4) in all SFP specimens shown in S3.1–S3.3 Fig, the second step of the stratified genotyping strategy was then applied using extension primer 067744–324 (*cf*. Table 2) to further analyze the extremely high-intensity Peak T shown in S3.1–S3.3 Fig, representing Genotypes #5–6 of AT-biased Cluster-A and Genotype #15 of AT-biased Cluster-B (*cf*. Fig 2).

Fig 5.1 shows a mass spectrum of extension primer 067744–324 (reverse complement) (*cf*. Table 2) for the pre-ejection SFP. Peak T possessing extremely high intensity represented Genotypes #6 and #15 (*cf*. Fig 1), while the component Genotype #4 was attributed only to a very small portion of Peak T, as demonstrated in S3.1 Fig. Peak C possessing the second highest intensity represented Genotype #5. Although an extended cytosine may be produced in the extension reaction on the DNA templates of Genotypes #16–17 using primer 067744–324 (*cf*. Fig 1), these 2 genotypes were absent in the *Cordyceps sinensis* SFP when using genotype-specific primers and PCR amplicon-cloning techniques [31]. Peaks C and T presented a C:T intensity ratio of 0.71 (Table 4). Peak G was absent. Peak A showed a low intensity, resulting in a C:A intensity ratio of 4.56 (Table 4), representing a co-occurring transversion point mutation genotype of unknown upstream and downstream sequences.

Fig 5.2 shows a mass spectrum for the post-ejection SFP, displaying a high intensity of Peak C (Genotype #5). Peak T, representing Genotypes #6 and #15, appeared in the mass spectrum as a shoulder of the acceding MS loop and was dramatically attenuated compared to that shown in Fig 5.1, resulting in a significantly elevated C:T ratio of 10.6 (Table 4). Peak G showed a low intensity, resulting in a C:G ratio of 53, while Peak A remained at a low level with an elevated C:A ratio of 7.57 (Table 4).

Fig 5.3 shows the overlapping mass spectra of Fig 5.1 and 5.2 after adjusting and aligning the vertical axis for intensity and visualizes the dynamic alteration of the allelic peaks in the SFPs from pre-ejection to post-ejection.

S4 Fig shows a mass spectrum for the SFP of developmental failure. The mass spectrum topology was similar to that shown in Fig 5.2, except that the intensity of Peak C (Genotype #5) was further elevated, resulting in a further elevated C:T ratio (Table 4). Peak G was nearly flat and negligible, while Peak A remained at a low intensity, with an elevated C:A ratio of 8.50 (Table 4).

In contrast to the consistently low levels of Genotype #4 in all 3 SFP samples shown in S3.1–S3.3 Fig, the second step of the stratified analysis indicated dramatic alterations of Peak T, which showed a very high level in the pre-ejection SFP (Fig 5.1) but presented significant attenuation in the SFPs after ejection (Fig 5.2) and of developmental failure (S4 Fig). Peak T represents the sum of intensities of Genotypes #6 and #15, as well as a very small portion of Genotype #4. Both Genotypes #4 and #15 of AT-biased Cluster-B were present in the *Cordyceps sinensis* SFP but absent in ascospore samples, as demonstrated by Li et al. [31, 49]. In contrast, Genotype #5 of AT-biased Cluster-A, which was represented by Peak C in Figs 5 and S4, remained at a consistently high intensity in all SFP samples. Transversion mutation genotypes of unknown upstream and downstream sequences either remained at a constantly low level (Peak A) or were absent (Peak G) in the SFP samples. Genotype #6 of AT-biased Cluster-A underwent dynamic quantitative alteration along with Genotype #15 of AT-biased Cluster-B when comparing Peak T in Fig 5.1 and 5.2. Genotype #6 was clustered phylogenetically in the clade of AT-biased Cluster-A, as shown in Fig 2, but it was closer to the clade of AT-biased

Cluster-B than other genotypes in AT-biased Cluster-A. The results may indicate that Genotype #6 is on an evolutionary branch different from other genotypes of AT-biased Clusters A and B (*cf*. Fig 2) and functions symbiotically in the *Cordyceps sinensis* SFPs.

## SNP genotyping analysis to distinguish between the GC- and AT-biased genotypes of *O. sinensis* in the fully and semi-ejected ascospores

Li et al. [31] reported the detection of Genotype #13 in semi-ejected ascospores but not in fully ejected ascospores and the detection of Genotype #14 in fully ejected ascospores but not in semi-ejected ascospores. These 2 offspring genotypes of *O. sinensis* were characterized by large DNA segment reciprocal substitutions and genetic material recombination between 2 parental fungi (Group A *H. sinensis* and an AB067719-type Group E fungus [7, 9–11, 31]). The ITS1-5.8S-ITS2 sequences of Genotypes #13–14 were alternatively segmentally inherited from their parental fungi [31]. The segment sequences of Genotypes #13–14 are either completely identical to or far different from those of Genotype #1, and the primer pairs for the first-step PCR listed in Table 1 are unfavorable for amplifying the ITS sequences of Genotypes #13–14. Thus, the genotyping study presented herein does not cover the analysis of GC-biased Genotypes #13–14.

6.1 shows a mass spectrum of extension primer 067721–477 (*cf*. Table 2) for fully ejected ascospores. Peak G represents GC-biased Genotype #1 (*cf*. Fig 1). Peak A represents transition point mutation AT-biased genotypes. Peak C was absent. Peak T had a very low intensity, representing a transversion point mutation genotype of unknown upstream and downstream sequences. The primers employed for the first-step PCR preferentially amplified the sequences of GC-biased genotypes, producing an amplicon pool containing sequences of GC-biased genotypes in greater quantities than the AT-biased sequences. The amplicon pool was used as the template for the second-step PCR, showing the higher-intensity Peak G and the lower-intensity Peak A. The G:A and G:T ratios were 2.5 and 15.0, respectively (Table 5).

Fig 6.2 shows a mass spectrum for the semi-ejected ascospores with a similar MS topology to that shown in Fig 6.1. Peak G represents GC-biased Genotype #1, and Peak A represents AT-biased genotypes. Peak C was absent, and Peak T had a very low intensity. There were no significant changes in the intensity ratios of G:A (2.2) and G:T (13.8) relative to those for the fully ejected ascospores (Table 5).

The extension primer 067721–531 (*cf*. Table 2) was also used to distinguish between the GC- and AT-biased genotypes of *O. sinensis* in the fully and semi-ejected ascospores. Mass spectra (not shown) showed the same topologies of Peaks A and G in both panels of Fig 6, except for the absence of both Peaks C and T representing transversion alleles. There was no meaningful change in the G:A intensity ratio (S5 Table).

## SNP genotyping analysis to distinguish between the AT-biased genotypes of *O. sinensis* in fully and semi-ejected ascospores

Li et al. [31] reported the absence of genotypes of AT-biased Cluster-B (Genotypes #4 and #15; *cf*. Fig 2) in the 2 types of ascospores. SNP genotyping using extension primer 067740–328 (reverse complement) (*cf*. Table 2) confirmed that both mass spectra (not shown) for the fully and semi-ejected ascospores showed a single Peak T of high intensity, representing Genotypes #5–6 and #16 of AT-biased Cluster-A (*cf*. Fig 2) [31]. Peaks C (Genotype #4 of AT-biased Cluster-B) and G (a transversion allele of unknown upstream and downstream sequences) are nearly flat. Peak A with very low intensities indicated the co-occurrence of a transversion mutation genotype of unknown upstream and downstream sequences in both ascosporic samples. The increase in the T:A ratio from 14.0 in fully ejected ascospores to 22.0 (S6 Table) in

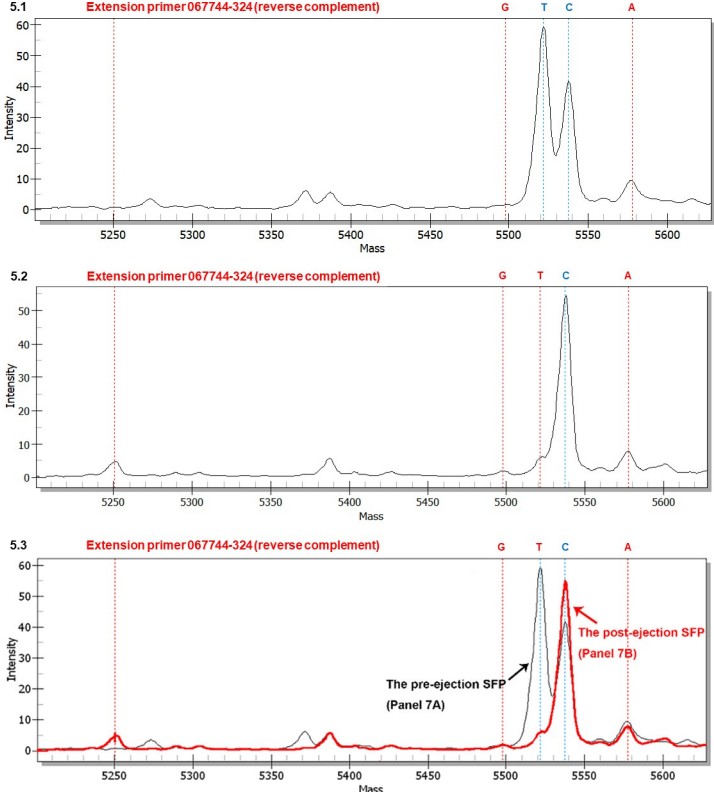

**Fig 5. MALDI-TOF mass spectra of extension primer 067744–324 (reverse complement) to distinguish between the AT-biased genotypes in the SFPs (with ascocarps) prior to and after ascospore ejection.** The extension reaction proceeded from extension primer 067744–324 (reverse complement) toward the SNP at position 324 in the AB067740 sequence (*cf*. Fig 1 for the location). In the allelic peaks, "G" represents the primer with an extended reverse complement cytosine (guanine in the sense chain; *cf*. Fig 1); "T" represents an extended reverse complement adenine (thymine in the sense chain); "C" denotes an extended reverse complement guanine (cytosine in sense chain); and "A" indicates an extended reverse complement thymine (adenine in the sense chain). Panel 5.1 shows the mass spectrum of the pre-ejection SFP. Panel 5.2 shows the mass spectrum of the post-ejection SFP. Panel 5.3 shows the overlapping mass spectra for Panels 5.1 (tracing in black) and 5.2 (tracing in red) with alignment of the horizontal and vertical axes of the panels.

semi-ejected ascospores was due to the considerably increased intensity of Peak T, the genotypes of AT-biased Cluster-A.

**Table 4. Mass intensity ratios of the SNP allelic peaks of transition and transversion mutant genotypes in SFPs prior to and after ascospore ejection and of developmental failure.**

| Extension primer | Allelic ratio | Intensity ratio | | |
|---|---|---|---|---|
| | | **Pre-ejection SFP (*cf*. Fig 5.1)** | **Post-ejection SFP (*cf*. Fig 5.2)** | **SFP of developmental failure (*cf*. S4 Fig)** |
| 067744–324 | C:T | 0.71 (41÷58) | 10.6 (↑↑↑; 53÷5.0) | 15.1 (↑↑↑↑; 68÷4.5) |
| | C:G | – | 53.0 (53÷1.0) | – |
| | C:A | 4.56 (41÷9.0) | 7.57 (↑; 53÷7.0) | 8.50 (↑; 68÷8.0) |

Note: Peak C in Figs 5 and S4 represents AT-biased Genotype #5 of *O. sinensis*, and Peak T represents AT-biased Genotypes #6 and #15 of *O. sinensis* and an extremely low amount of Genotype #4. Peaks G and A denote 2 transversion mutation genotypes of unknown upstream and downstream sequences. "↑" (increase less than two-fold) and "↑↑↑↑" (increase greater than four-fold) denote significant increases in intensity ratios compared to that for the pre-ejection SFP. "–" means that one of the allelic peaks was missing, and no ratio could be calculated.

Fig 7.1 shows a mass spectrum obtained using extension primer 067740–324 (*cf.* Table 2) from the fully ejected ascospores. Peak C represented AT-biased Genotype #5 (*cf.* Fig 1), while AT-biased Genotype #17, with the same extended cytosine, was absent in the ascospores based on the use of genotype-specific primers and cloning-based amplicon sequencing [31]. Peak T represented AT-biased Genotypes #6 and #16 of AT-biased Cluster-A, while Genotypes #4 and #15 of AT-biased Cluster-B, with the same extended thymine, were absent in the ascospores [31]. The lower intensity of Peak C and the higher intensity of Peak T resulted in a C:T ratio of 0.69 (Table 6). Peaks A and G represent extremely low-intensity transversion point mutation genotypes of unknown upstream and downstream sequences, with C:A and C:G ratios of 9.06 and 15.4, respectively (Table 6).

Fig 7.2 shows a mass spectrum for semi-ejected ascospores. The intensity of Peak C (Genotype #5) was elevated more than two-fold, from 15.4 in fully ejected ascospores to 34.8 in semi-ejected ascospores, as visualized in Fig 7.3 after adjusting and aligning the vertical axis for intensities for the overlapping mass spectra of Fig 7.1 and 7.2. Peak T (Genotypes #6 and #16) was slightly elevated from 22.3 to 26.8 (Fig 7.1 and 7.2). The asynchronous, disproportional increases in intensity (Fig 7.3) resulted in an elevation of the C:T ratio from 0.69 to 1.30 from the fully ejected ascospores to the semi-ejected ascospores (Table 6). The intensities of Peaks A and G were slightly increased. The C:A and C:G intensity ratios were increased from 9.06 to 14.5 and from 15.4 to 18.3, respectively (Table 6).

## Discussion

### Biochip-based MassARRAY MS SNP genotyping technique for semi-quantitative analysis

Natural *Cordyceps sinensis* is an integrated microecosystem of the insect-fungal complex [6, 7], and its fungal members are differentially alternated in different combinations and play different biological roles in its complex lifecycle (personal communication with Prof. Liang Z-Q). Along with >90 fungal species from at least 37 genera [15, 16, 30], 17 genotypes of *O. sinensis* of genetically and phylogenetically distinct (*cf.* Figs 1 and 2) have been identified, which differentially co-occur in the stroma and caterpillar body [6, 7, 13, 19–23, 48], SFP (with ascocarps) and ascospores of natural *Cordyceps sinensis* [31, 49] or in entire *Cordyceps sinensis* [36, 37, 47], as previously summarized [6–11]. The nature of *Cordyceps sinensis* causes technical difficulties and debates in detection of fungal members and in monitoring their dynamic changes during development and maturation of the natural product.

The stroma and caterpillar body of *Cordyceps sinensis* have been examined using the Southern blotting technique, and the natural abundance (without DNA amplification) of GC- and AT-biased genotypes of *O. sinensis* have been found to exhibit dynamic alterations in an asynchronized and disproportional manner in the compartments of natural *Cordyceps sinensis* during maturation [19, 23]. Although Southern blotting analysis is not believed to be supersensitive, it is able to demonstrate the natural abundance of DNA species in test materials, reflecting the natural biomasses of the target DNA moieties. However, the Southern blotting technique can only detect *O. sinensis* genotypes as GC- and AT-biased groups after the samples are pre-treated with endonucleases [19, 23], not individually. On the contrary, PCR-based techniques are considered sensitive molecular approaches, they may lead to disproportional exponential amplifications and inaccurate quantifications of various metagenomic components in complex DNA samples because of the involvement of multiple technical factors influencing the PCR process and sometimes cause failure of DNA amplification and sequencing for various reasons [8]. Restriction fragment length polymorphism (RFLP) techniques combined with cloning-based amplicon sequencing has also been employed to monitor the

**Table 5. Mass intensity ratios of the SNP allelic peaks of transition and transversion mutant genotypes in fully and semi-ejected ascospores.**

| Extension primer | Allelic ratio | Intensity ratio | |
|---|---|---|---|
| | | Fully ejected ascospores | Semi-ejected ascospores |
| 067721–477 | | (*cf.* Fig 6.1) | (*cf.* Fig 6.2) |
| | G:A | **2.5** (7.5÷3.0) | **2.2** (↔; 6.7÷3.2) |
| | G:C | − | − |
| | G:T | **15.0** (7.5÷0.5) | **13.8** (↔; 6.9÷0.5) |

Note: Peak G represents GC-biased Genotype #1 *H. sinensis*; Peak A indicates AT-biased genotypes of *O. sinensis* (*cf.* Fig 1). Peaks C and T denote 2 transversion mutation genotypes of unknown upstream and downstream sequences. "↔" indicates no significant change (within 20% variation) compared to the intensity ratios for fully ejected ascospores. "−" means that one of the allelic peaks was missing, and no ratio could be calculated.

dynamic changes of GC-biased Genotypes #1–2 of *O. sinensis* during *Cordyceps sinensis* maturation [19, 20]. This technique has limitations in quantitative analysis of multiple *O. sinensis* genotypes. Technical difficulties are encountered when using quantitative real-time PCR (qrPCR) to quantify and monitor the dynamic changes of metagenomic members in multi-fungal natural products, such as *Cordyceps sinensis*, because considerable sequence differences of fungi prevent balanced design of primers and probes for unbiased amplifications of the sequences of target intrinsic fungi. The current study used biochip-based MassARRAY MOL-DI-TOF mass spectrometry SNP genotyping to examine teleomorphic genotypes of *O. sinensis* and monitor the dynamic alterations of the individual genotypes in the SFPs and ascospores of natural *Cordyceps sinensis*. This technique was used previously to monitor the dynamic alterations of coexisting *O. sinensis* genotypes in the stromata and caterpillar bodies of *Cordyceps sinensis* during maturation [19, 21–23].

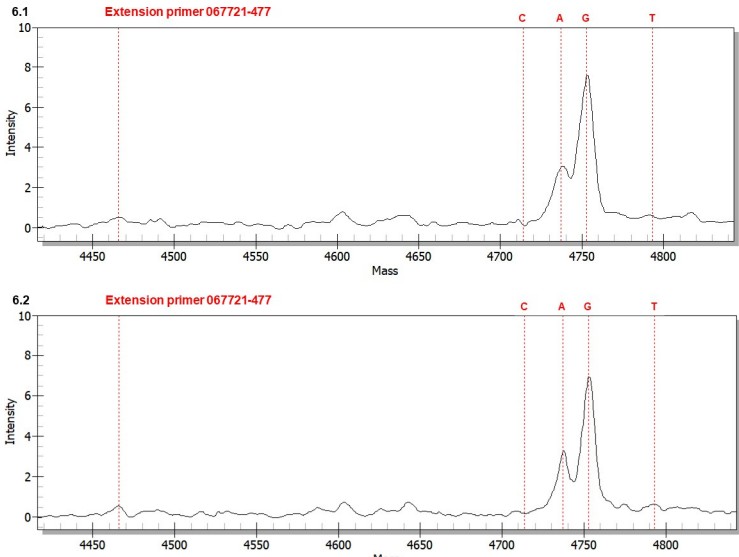

**Fig 6. MALDI-TOF mass spectra of extension primer 067721–477 to distinguish between GC- and AT-genotypes in fully and semi-ejected ascospores.** The extension reaction proceeded from extension primer 067721–477 toward the SNP at position 477 in the AB067721 sequence (*cf.* Fig 1 for the location). In the allelic peaks, "C" denotes the primer with an extended cytosine; "A" indicates an extended adenine; "G" represents an extended guanine; and "T" represents an extended thymine. Panel 6.1 shows the mass spectrum of fully ejected ascospores. Panel 6.2 shows the mass spectrum of semi-ejected ascospores.

We designed two types of PCR-based experiments in this research project: (1) qualitative examinations using multiple sets of species- and genotype-specific primers for touch-down PCR and cloning-based amplicon sequencing with collection and sequencing of more than 30 white colonies to profile the genotypic components of the metagenomic DNA extracted from the immature and mature stromata, SFPs with ascocarps, and 2 types of ascospores of natural *Cordyceps sinensis*, as presented in the companion paper [31]; and (2) a semiquantitative analysis using SNP mass spectrometry genotyping to monitor the dynamic alterations of the various genotypes in the SFPs densely covered with numerous ascocarps prior to and after ascospore ejection and of developmental failure and in the 2 types of ascospores, as presented herein.

Secondary structures/conformation in ITS1-5.8S-ITS2 sequences of *O. sinensis* genotypes have been recognized previously and greatly impact and even perplex PCR amplification and amplicon sequencing, often causing false negative results and consequently incorrect conclusions [8–11, 18, 49]. MassARRAY SNP MALDI-TOF MS genotyping represents a useful technique to overcome the shortcomings of regular PCR identification [54]. It is a two-step PCR-mass spectrometry technique involving a 384-well biochip [19–22, 54, 58]. Sequences containing various SNP alleles of different genotypes may not be amplified at the same rate because of (1) different amounts of DNA templates (*i.e.*, the amplicons obtained from the first-step PCR) in the samples; (2) variable DNA sequences and related secondary structures; (3) different primer binding rates and DNA chain extension rates, causing the MS intensities of the SNP alleles to not directly reflect the copy numbers of the genotypic sequences; and (4) perhaps other reasons in the natural world [8]. However, under the same PCR conditions using the same pair of primers in the first-step PCR, the same extension primer in the second-step PCR, and the same batch of reaction buffer run according to the same PCR settings and extension reaction on a single 384-well chip, the sequences of genotypes will be amplified at comparable amplification rates from one reaction well to another in a single chip.

Although direct quantitative comparisons of the intensities of the same allelic peaks in different mass spectra may not be accurate, it is reasonable to select the intensity of one SNP allelic peak in the mass spectra as the internal reference for intensity ratio computation (Tables 3–6 and S3–S6), which provides legitimate information for semiquantitative comparisons of the relative abundances of genotypes in different mass spectra and semiquantitative monitoring of the alterations of the genotypes among the study samples. Such semiquantitative comparisons of intensity ratios are logical and valid within a systematic error, reflecting the relative alterations of GC- and AT-biased genotypes of *O. sinensis* and the transversion mutation genotypes of unknown upstream and downstream sequences in the SFPs prior to and after ascospore ejection and of developmental failure and in fully and semi-ejected ascospores collected from the same *Cordyceps sinensis* specimens.

**Dynamic alterations of the genotypes of *O. sinensis*.** Southern blotting, restriction fragment length polymorphism (RFLP) analysis combined with amplicon sequencing with or without amplicon cloning, SNP genotyping, and other molecular examinations demonstrated that GC-biased Genotype #1 *H. sinensis* was never the dominant fungal species in the *Cordyceps sinensis* stroma and that AT-biased genotypes were the dominant species [7, 9–11, 19–23, 31]. Li et al. [49] proposed that all AT-biased genotypes are "ITS pseudogenes . . . in a single genome" of Genotype #1 *H. sinensis*. However, the 5 genome assemblies (ANOV00000000, JAAVMX000000000, LKHE00000000, LWBQ00000000, and NGJJ00000000) of *H. sinensis* Strains Co18, IOZ07, 1229, ZJB12195, and CC1406-203, respectively [41–45], do not contain the sequences of Genotypes #2–17, which belong to the genomes of independent *O. sinensis* fungi [6–11, 13, 19–23, 31]. Multiple repetitive ITS copies were identified in the genome assemblies JAAVMX000000000 and NGJJ00000000 of Genotype #1 *H. sinensis* Strains IOZ07 and CC1406-203, respectively (*cf.* Fig 2) [44, 45]. These multiple genomic ITS repeats are GC-

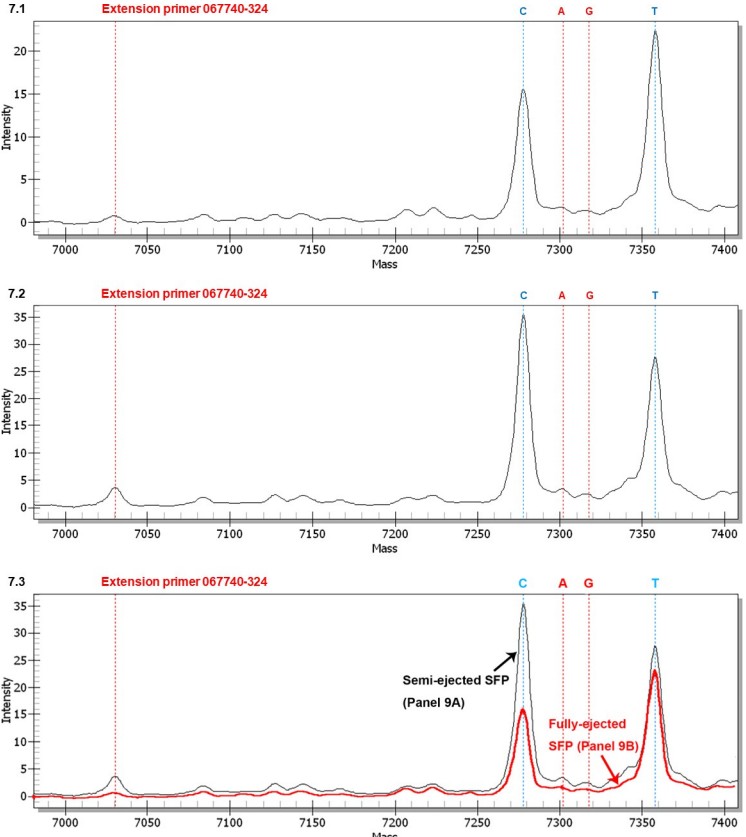

**Fig 7. MALDI-TOF mass spectra of extension primer 067740–324 to distinguish between AT-biased genotypes in the fully and semi-ejected ascospores.** The extension reaction proceeded from extension primer 067740–324 toward the SNP at position 324 in the AB067740 sequence (*cf*. Fig 1 for the location). In the allelic peaks, "C" denotes the primer with an extended cytosine; "A" indicates an extended adenine; "G" represents an extended guanine; and "T" indicates an extended thymine. Panel 7.1 shows the mass spectrum of fully ejected ascospores. Panel 7.2 shows the mass spectrum of semi-ejected ascospores. Panel 7.3 shows the overlapping mass spectra for Panels 7.1 (black tracing) and 7.2 (red tracing) with alignment of the horizontal and vertical axes of the panels.

biased and feature multiple scattered insertion/deletion and transversion point mutations, which might not be induced through RIP mutagenesis that theoretically causes cytosine-to-thymine transition mutations [11, 57]. The ML phylogenetic tree shown in Fig 2 of the current

**Table 6. Mass intensity ratios for the SNP allelic peaks for multiple transition and transversion mutation genotypes in the fully and semi-ejected ascospores.**

| Extension primer | Allelic ratio | Intensity ratio | |
|---|---|---|---|
| | | **Fully ejected ascospores (*cf*. Fig 7.1)** | **Semi-ejected ascospores (*cf*. Fig 7.2)** |
| 067740–324 | C:T | **0.69** (15.4÷22.3) | **1.30** (↑; 34.8÷26.8) |
| | C:A | **9.06** (15.4÷1.7) | **14.5** (↑; 34.8÷2.4) |
| | C:G | **15.4** (15.4÷1.0) | **18.3** (↑; 34.8÷1.9) |

Note: In the 067740–328 primer section (upper panel), Peak T represents AT-biased Genotype #4 of *O. sinensis*, and Peak T indicates other AT-biased genotypes of *O. sinensis* (*cf*. Fig 1). In the 067740–324 primer section (lower panel), Peak C represents AT-biased Genotype #5 of *O. sinensis*, and Peak T indicates other AT-biased genotypes of *O. sinensis* (*cf*. Fig 1). Peaks G and A in both panels denote 2 transversion mutation genotypes of unknown upstream and downstream sequences. "↑" indicates a significant increase (less than two-fold) in the intensity ratio compared to that of the fully ejected ascospores. "—" means missing one of the allele peaks, and no ratio could be calculated.

paper and a Bayesian majority rule consensus tree shown in Fig 2 of [57] demonstrated that all sequences of repetitive ITS copies in JAAVMX000000000 and NGJJ00000000 were clustered into the GC-biased genotype clade along with GC-biased genotypes, far distant from the AT-biased genotype clade. The dynamic alterations of GC- and AT-biased genotypes of *O. sinensis* in an asynchronous, disproportional manner observed in this study (*cf.* Figs 4–2 and S1–S4) and previous studies [19–23] further confirmed the genomic independence of the multiple genotypes, providing evidence of the coexistence of interindividual *O. sinensis* fungi [13, 19–23]. These results thus disproved the hypothesis of "ITS pseudogenes . . . in a single genome", which was based on the misinterpretation of genomic residency data for the multiple AT-biased mutant genotypes in the genome of GC-biased Genotype #1 *H. sinensis* [6–11, 13, 31].

In addition to the companion paper [31] that reported the differential co-occurrence of multiple genotypes of *O. sinensis* in different combinations in the immature and mature stroma, SFPs with ascocarps, and ascospores of natural *Cordyceps sinensis* using qualitative molecular techniques, the current paper also revealed the dynamic alterations of GC- and AT-biased genotypes in the SFPs (with ascocarps) and ascospores through a semiquantitative SNP genotyping approach (*cf.* Figs 4–2 and S1–S4). For instance, S2.1 Fig demonstrates the nearly identical intensities of Peaks A and G, representing AT- and GC- biased genotypes of *O. sinensis*, in the pre-ejection SFP, and S2.2 Fig shows a significant reduction of Peak A (AT-biased genotypes) after ascospore ejection, resulting in a dramatic increase in the G:A intensity ratio (*cf.* S3 Table). Fig 7.1–7.3 demonstrate that Peak C (AT-biased Genotype #5) with slightly lower intensity (than that for Peak T) in the fully ejected ascospores increased more than two-fold in its peak height in the semi-ejected ascospores collected from the same pieces of natural *Cordyceps sinensis* specimens, compared with Peak T (AT-biased Genotypes #6 and #16) with very high intensity in the fully ejected ascospores but only a disproportional minimal increase in the semi-ejected ascospores (*cf.* Fig 7 and Table 6). Genotypes #5–6 and #16 were phylogenetically clustered into AT-biased Cluster-A (*cf.* Fig 2). Significant changes in the intensities of AT-biased genotypes are also shown in other figures and tables, indicating that the sequences of AT-biased genotypes belong to the independent genomes of interindividual *O. sinensis* fungi.

The dynamic alterations of the intensities of the allelic peaks suggested the genomic independence of GC- and AT-biased *O. sinensis* genotypes, inconsistent with the hypothesis that multiple AT-biased "ITS pseudogene" components of *O. sinensis* belong to "a single genome" of Genotype #1 *H. sinensis* [49]. Notably, the intensities of the allelic peaks and their alterations shown in Figs 4 and 6 and S1 and S2 do not necessarily reflect the natural biomasses of the GC- and AT-biased genotypes of *O. sinensis* because the primers employed for the first-step PCR preferentially amplified GC-biased genotypes over AT-biased genotypes, producing a greater amount of GC-biased amplicons that served as the templates for primer extension reactions in the second-step PCR.

Mass spectra obtained using extension primers 067721–447 and 067721–531 demonstrated the co-occurrence of GC- and AT-biased *O. sinensis* genotypes in fully and semi-ejected ascospores, resulting in essentially unchanged G:A ratios (*cf.* Fig 6 and Tables 5 and S5).

GC-biased Genotype #1 *H. sinensis* is postulated to be an anamorph of *O. sinensis* [32], although it may not be the sole anamorph [11, 31, 49, 59, 60]. In addition, GC-biased Genotype #1 was reported to be the sole teleomorph of *O. sinensis* in natural *Cordyceps sinensis* [32, 40], and AT-biased Genotype #4 was the sole teleomorph of *O. sinensis* in cultivated *Cordyceps sinensis* [40]. Our findings presented in this paper and the companion paper [31] indicate that multiple co-occurring teleomorphic AT-biased genotypes may play critical symbiotic roles in the development and maturation of SFPs with ascocarps and ascospores in the sexual life stage of *Cordyceps sinensis*. Without certain AT-biased *O. sinensis* genotypes in various

combinations, ascocarps may not be able to normally develop and mature to support the development, maturation, and ejection of the *Cordyceps sinensis* ascospores (*cf.* Figs 3–5 and S1–S4), especially lacking high intensities of Genotype #6 (AT-biased Cluster-A) and Genotype #15 (AT-biased Cluster-B) in the SFP of developmental failure, as shown in S4 Fig. We also noticed the coexistence of transversion mutated alleles of unknown upstream and downstream sequences that were altered in the SFPs and ascospores (Figs 4–2 and S1–S4 and Tables 3–6 and S4 and S6), suggesting that these co-occurring transversion genotypes may also participate symbiotically in the development and maturation of ascocarps and ascospores.

**Stratification strategy for genotyping AT-biased genotypes.** The stratification genotyping strategy used in this study profiled in a step-by-step manner the multiple genotypic components of AT-biased Clusters A and B in the *Cordyceps sinensis* SFPs. The first step of stratifying assays used extension primer 067740–328. The mass spectra of the SFPs prior to and after ascospore ejection and of developmental failure demonstrated the existence of an extremely small quantity of Genotype #4 of AT-biased Cluster-B that is represented by Peak C in S3.1–S3.3 Fig, in addition to large amounts of Genotypes #5–6 of AT-biased Cluster-A and Genotype #15 of AT-biased Cluster-B. In contrast to the dominance of Genotype #4 of AT-biased Cluster-B in the stroma of immature *Cordyceps sinensis* [21, 22], in its asexual growth stage, the abundance of Genotype #4 dramatically declined in the stroma of mature *Cordyceps sinensis* [19, 21, 22] and retained a very low level in the SFPs (with ascocarps) in the sexual stage of the *Cordyceps sinensis* lifecycle [31].

The second step of the stratification analysis involved the extension primer 067744–324. The mass spectrum (*cf.* Fig 5.1) demonstrated a higher intensity of Peak T (sum of a minimal quantity of Genotype #4 and a large amount of Genotypes #6 and #15) and a slightly lower intensity of Peak C (Genotype #5) in the pre-ejection SFP, resulting in a C:T ratio of 0.71 (*cf.* Table 4). In contrast to retaining its high intensity of Peak C (Genotype #5 of AT-biased Cluster-A) in the post-ejection SFP, Peak T was significantly attenuated in the SFPs after ascospore ejection (*cf.* Fig 5.2), representing a dramatic reduction in Genotype #6 (AT-biased Cluster-A) and Genotype #15 (AT-biased Cluster-B) along with the extremely small quantity of component Genotype #4 in the *Cordyceps sinensis* SFP after ascospore ejection; this caused a significant increase in the C:T ratio (*cf.* Table 4). Peak T (Genotypes #6 and #15, and minimal amount of Genotype #4) also showed extremely low intensity in the SFP of developmental failure (*cf.* S4 Fig), resulting in an extremely high C:T ratio (*cf.* Table 4). These findings indicated that AT-biased Genotypes #6 and #15 may play critical roles in the development and maturation of *Cordyceps sinensis* ascocarps and that insufficient Genotypes #6 and #15 in the SFP may cause developmental failure of *Cordyceps sinensis* ascocarps (comparing Figs 5.1 and S4), while AT-biased Genotype #4 may play an important role in the early asexual developmental stage of natural *Cordyceps sinensis* [21–22, 31]. Although both Genotypes #4 and #15 were phylogenetically clustered into AT-biased Cluster-B, they play distinct symbiotic roles in different maturation stages of *Cordyceps sinensis*.

Genotype #6 was clustered into AT-biased Cluster-A, but its phylogenetic leaf was closer to Cluster-B than other Cluster-A genotypes (*cf.* Fig 2), indicating that Genotype #6 may play different symbiotic roles from Genotypes #5 and #16 of AT-biased Cluster-A. We also noticed that all Genotype #6 sequences uploaded in GenBank are short, missing small segments at the 5' end of ITS1 and the 3' end of ITS2, indicating that unique secondary conformation may exist in these segments that prevents PCR detection of Genotype #6 when using the so called "universal" primers (primers *ITS1–ITS5*) and that design of specific primers may need to detect Genotype #6 sequence.

Genotypes #4 and #15 of AT-biased Cluster-B were absent in the ascospores of *Cordyceps sinensis* [31, 49], although Wei et al. [40] reported that Genotype #4 was the sole teleomorph of

*O. sinensis* in cultivated *Cordyceps sinensis*. Genotype #5 (Peak C in Fig 7) and Genotypes #6 and #16 (components of Peak T in Fig 7) were phylogenetically clustered into AT-biased Cluster-A (*cf.* Fig 2) and play symbiotic roles in assisting the function of GC-biased Genotype #1 *H. sinensis* in the sexual life stage of natural *Cordyceps sinensis*. Fungal symbiosis is critical for the development, maturation, and ejection of *Cordyceps sinensis* ascospores.

This paper and its companion paper [31] demonstrated the symbiotic co-occurrence of GC-biased Genotype #1 and AT-biased Genotypes #5–6 and #16 of *O. sinensis* in *Cordyceps sinensis* ascospores. However, Li et al. [49] reported the detection of GC-biased Genotype #1 and AT-biased Genotype #5 in 8 of 15 clones in 25-day cultures generated from mono-ascospores. Apparently, the culture-dependent study by Li et al. [49] overlooked AT-biased Genotypes #6 and #16 as well as GC-biased Genotype #14 in fully ejected ascospores [31], which might be attributed to either nonculturability of these overlooked genotypes of *O. sinensis* fungi under the 25-day *in vitro* culture conditions set by [49] or a failure of detection due to an inappropriate design of the molecular methodology. As shown in Fig 1, the sequences of AT-biased Genotypes #6 (EU555436) and #16 (KT232019) are both short compared with that of Genotype #1 *H. sinensis*; Genotype #6 EU555436 is especially short, lacking short DNA segments close to its 5' and 3' ends. There have been no reports of the detection of Genotypes #6 and #16 using "universal" primers. The molecular features (secondary structure/conformation) of these genotypes may prevent their detection through PCR when using "universal" primers [6–11, 19, 21].

Li et al. [31] reported the detection of Genotype #13 in semi-ejected ascospores and Genotype #14 in fully ejected ascospores. These 2 GC-biased genetically variant genotypes feature large DNA segment reciprocal substitutions and genetic material recombination between the 2 parental fungi (Group-A Genotype #1 *H. sinensis* and a Group-E AB067719-type fungus), regardless of whether Genotypes #13–14 are the offspring derived from fungal hybridization or parasexual reproduction [7, 11, 59]. Using genotype-specific primers and cloning-based amplicon sequencing techniques is an appropriate approach to identify these 2 offspring genotypes, as well as other AT-biased genotypes [31]. However, the MassARRAY SNP mass spectrometry genotyping technique using the primer pairs and extension primers designed in this study is not applicable to identify ascosporic Genotypes #13–14 because the ITS1-5.8S-ITS2 sequences of Genotypes #13–14 were alternatively segmentally inherited from their parental fungi [31], either completely identical to or far different from those of Genotype #1 and the primer pairs for the first-step PCR (*cf.* Fig 1 and Table 1). Thus, the semiquantitative study presented in this paper does not cover the analysis of GC-biased Genotypes #13–14.

*True causative fungus/fungi of natural Cordyceps sinensis*. Zhang et al. [61] summarized nearly 40 years of mycological experience involving the failed induction of the fruiting body and ascospore production in research-oriented laboratory settings, either in fungal cultures or after infecting insects with fungal inoculants. Hu et al. [41] reported success in inducing infection-mortality-mummification in 40 larvae of *Hepialus* sp. in an experiment using a mycelial mixture of anamorphic *H. sinensis* Strains Co18 and QH195-2 as inoculants but failed to induce the production of fruiting bodies.

Li et al. [58] demonstrated that a mixture of wild-type Strains CH1 and CH2 induced a favorable infection-mortality-mummification rate of 55.2±4.4% in 100 larvae of *Hepialus armoricanus*, which represents a 15–39-fold greater inoculation potency (P<0.001) than the rates (1.4–3.5%) achieved after inoculation with the conidia or mycelia of *H. sinensis* or pure ascospores of *Cordyceps sinensis* (n = 100 larvae in each inoculation experiment). The wild-type CH1 and CH2 strains exhibit *H. sinensis*-like morphological and growth characteristics but contain GC-biased Genotype #1 *H. sinensis*, AT-biased Genotypes #4–6 of *O. sinensis*, and *Paecilomyces hepiali*.

Wei et al. [40] reported an industrial artificial inoculation project and demonstrated a species contradiction between the anamorphic inoculant GC-biased Genotype #1 *H. sinensis* (Strains 130508-KD-2B, 20110514 and H01-20140924-03) and the sole teleomorph of AT-biased Genotype #4 of *O. sinensis* in cultivated *Cordyceps sinensis*. In addition to reporting the sole teleomorph of AT-biased Genotype #4 in cultivated *Cordyceps sinensis*, Wei et al. [40] also reported the sole teleomorph of GC-biased Genotype #1 in natural *Cordyceps sinensis*, reporting at least 2 teleomorphs of *O. sinensis*.

We reported the co-occurrence and dynamic alterations of multiple genotypes of *O. sinensis* in the teleomorphic SFPs and ascospores of natural *Cordyceps sinensis* in this study and the companion study [31], suggesting that Genotype #1 is not the sole teleomorph of *O. sinensis* and that Genotype #1 *H. sinensis* may be among the true causative fungi for natural and cultivated *Cordyceps sinensis*. Our findings of multiple teleomorphs of *O. sinensis* may explain the 40-year failure of induction of the fruiting body and ascospore production in research-orientated academic settings, as summarized by Zhang et al. [61]. The evidence from the studies [31, 40, 41, 58, 61] suggested that the synergistic inoculation of multiple fungal species is required to initiate the development of the stromal primordia and fruiting bodies of natural and cultivated *Cordyceps sinensis* and to subsequently achieve the biological transition from the initial asexual growth to sexual reproduction in the later maturational stages and complete the *O. sinensis* infection cycle and the natural *Cordyceps sinensis* lifecycle.

**Inability of self-fertilization reproductive behavior in *H. sinensis*.** The reproductive behavior of ascomycetes is controlled by transcription factors encoded at the mating-type (*MAT*) locus [60–63]. The expression of mating-type genes is controlled at the genomic, epigenetic, transcriptomic, post-transcriptive, translational, post-translational, and proteomic levels [59]. Zhang & Zhang [64] reported the differential occurrence of mating-type genes of *MAT1-1* and *MAT1-2* idiomorphs in many *Cordyceps sinensis* specimens and proposed a facultative hybridization hypothesis for *O. sinensis*. Li et al. [59] analyzed transcriptome assemblies of *H. sinensis* strains focusing on the mating-type genes of *MAT1-1* and *MAT1-2* idiomorphs and reported the differential transcription of mating-type genes in Genotype #1 *H. sinensis* strains, in addition to the differential occurrence of the mating-type genes in 237 *H. sinensis* strains.

Liu et al. [65] reported the transcriptome assembly GCQL00000000 of the *H. sinensis* Strain L0106, in which the MAT1-1-1 transcript was absent, but the MAT1-2-1 transcript was present [59]. Although Bushley et al. [50] reported the detection of transcripts of mating-type genes of both *MAT1-1* and *MAT1-2* idiomorphs in *H. sinensis* Strain 1229, the MAT1-2-1 transcript contained nonspliced intron I that contains 3 stop codons. The MAT1-2-1 transcript would then produce a largely truncated protein, lacking the major portion of the protein encoded by exons II and III of MAT1-2-1 gene [59], inevitably resulting in biological dysfunction of the truncated MAT1-2-1 protein. Thus, the evidence of the differential transcription of mating-type genes indicated the self-sterilization of Genotype #1 *H. sinensis* [59] and the inability to carry out self-fertilization under homothallic or pseudohomothallic reproduction [41, 50].

The evidence of genetic, transcriptional, and coupled transcriptional-translational controls of *O. sinensis* sexual reproduction indicated the requirement of mating partners to carry out sexual reproduction in the lifecycle of natural *Cordyceps sinensis*. The potential sexual partners in physiological heterothallism may be coexisting intraspecific GC-biased Genotypes #1 of *O. sinensis*, monoecious or dioecious [11, 59]. For instance, *H. sinensis* Strains 1229 and L0106 may be able to produce complementary mating proteins and potentially act as mating partners in physiological heterothallic reproduction [11, 41, 59, 65].

The potential mating partners may also be heterospecific fungal species co-colonized in natural and cultivated *Cordyceps sinensis* if the fungi are able to break through the interspecific reproductive isolation barrier to achieve hybridization. Notably, the taxonomic positions of

GC-biased Genotypes #2–3 and #7–14 and AT-biased Genotypes #4–6 and #15–17 have not been determined, probably due to their *H. sinensis*-like morphological and growth characteristics and nonculturability *in vitro* leading to technical difficulties in fungal isolation and purification [7, 9–11, 37, 47]. However, they most likely belong to independent *O. sinensis* fungi because their sequences do not reside in the genome of Genotype #1 *H. sinensis* [6–11, 41–46] and because of the phylogenetic analysis and conclusion by Stensrud et al. [55] that a large sequence variation in the conservative 5.8S gene "far exceeds what is normally observed in fungi . . . even at higher taxonomic levels (genera and family)." Studies have reported a close association of GC-biased Genotypes #1 and #2 and AT-biased genotypes of *O. sinensis* with dynamic genotypic alterations in the stroma of natural *Cordyceps sinensis* [5–11, 19–23], especially in the SFPs and ascospores as coexisting teleomorphs in the sexual life stages presented in this study and other studies [31, 49]. Hopefully, the multiple genotypes of *O. sinensis* reported in this study and the companion study [31] may potentially serve as sexual partners for hybridization if they are finally determined to be independent fungal species. Coincidentally, transcriptome assemblies of *H. sinensis* Strain L0106 and *P. hepiali* Strain Feng showed differential transcription of mating-type genes that were complementary to each other [59, 65, 66]. Barseghyan et al. [67] reported that *H. sinensis* and *Tolypocladium sinensis* were anamorphs of *O. sinensis*; however, the mating-type genes and their transcripts for *T. sinensis* have not been documented.

These findings provide genetic and transcriptomic evidence for the possible outcrossing of *O. sinensis* fungi and encourage further investigation of their reproductive physiology. In addition, Li et al. [7, 11, 31, 59] provided evidence of the potential hybridization or parasexual reproduction of Group-A *H. sinensis* (Genotype #1 of *O. sinensis*) and a Group-E AB067719-type fungus in *Cordyceps sinensis* ascospores, producing genetically variant offspring, Genotypes #13–14 of *O. sinensis*, each in the semi- or fully ejected ascospores. The evidence of complex reproduction behaviors of *O. sinensis* supports the facultative hybridization hypothesis for *O. sinensis* proposed by Zhang & Zhang [64].

## Conclusions

We have demonstrated in this paper the histology of the SFPs (densely covered with numerous ascocarps) prior to and after ascospore ejection and of developmental failure and differential cooccurrence and dynamic alterations of multiple teleomorphic genotypes of *O. sinensis* in different combinations in the SFPs and ascospores of natural *Cordyceps sinensis*. These findings in regard to the genital cells (ascospores) and their production organ (the ascocarps that densely cover the SFPs) provided evidence of the genomic independence of the *O. sinensis* genotypes, which do not reside in a single genome of Genotype #1 *H. sinensis* but instead belong to interindividual fungi. Metagenome members, including several genotypes of *O. sinensis*, play symbiotic roles in various developmental and maturational stages in different compartments of natural *Cordyceps sinensis* to accomplish sexual reproduction of *O. sinensis* during the lifecycle of natural *Cordyceps sinensis*.

## Supporting information

**S1 Fig. MALDI-TOF mass spectra of extension primer 067721–477 to distinguish between GC- and AT-biased genotypes in the SFP (with ascocarps) of developmental failure.** The extension reaction proceeded from extension primer 067721–477 toward the SNP at position 477 in the AB067721 sequence (*cf.* Fig 1 for the location). In the allelic peaks, "C" denotes extension of the primer with an extended cytosine; "A" indicates an extended adenine, "G"

represents an extended guanine, and "T" refers to an extended thymine.
(TIF)

**S2 Fig. MALDI-TOF mass spectra of extension primer 067721–531 to distinguish between the GC- and AT-genotypes in the SFPs (with ascocarps) before and after ascospore ejection and the SFP that failed to develop and eject the ascospores.** The extension primer 067721–531 was extended to the SNP at position 531 in the AB067721 sequence (*cf*. Fig 1 for the location). The allele peaks are marked: "C" denotes the primer with an extended cytosine; "A" indicates an extended adenine, "G" represents an extended guanine, and "T" refers to an extended thymine. Panel S2.1 shows the mass spectrum for the SFP before ascospore ejection. Panel S2.2 shows the mass spectrum for the SFP after ascospore ejection. Panel S2.3 shows the mass spectrum for the SFP that failed to develop and eject the ascospore. Panel S2.4 shows the overlapping mass spectra for Panels S2.1 (black tracing) and S2.2 (red tracing) with alignment of the horizontal and vertical axes of the panels.
(TIF)

**S3 Fig. MALDI-TOF mass spectra of extension primer 067740–328 (reverse complement) to distinguish between the AT-genotypes in the SFPs (with ascocarps) prior to and after ascospore ejection and the SFP of developmental failure.** The extension reaction proceeded from extension primer 067740–328 (reverse complement) toward the SNP at position 328 in the AB067740 sequence (*cf*. Fig 1 for the location). In the allelic peaks, "G" represents the primer with an extended reverse complement cytosine (guanine in the sense chain; *cf*. Fig 1); "T" represents an extended reverse complement adenine (thymine in the sense chain); "C" denotes an extended reverse complement guanine (cytosine in the sense chain); and "A" indicates an extended reverse complement thymine (adenine in the sense chain). Panel S3.1 shows the mass spectrum of the pre-ejection SFP. Panel S3.2 shows the mass spectrum of the post-ejection SFP. Panel S3.3 shows the mass spectrum of the SFP of developmental failure.
(TIF)

**S4 Fig. MALDI-TOF mass spectra of extension primer 067744–324 (reverse complement) to distinguish between the AT-biased genotypes in the SFP (with ascocarps) of developmental failure.** The extension reaction proceeded from extension primer 067744–324 (reverse complement) toward the SNP at position 324 in the AB067740 sequence (*cf*. Fig 1 for the location). In the allelic peaks, "G" represents the primer with an extended reverse complement cytosine (guanine in the sense chain; *cf*. Fig 1); "T" represents an extended reverse complement adenine (thymine in the sense chain); "C" denotes an extended reverse complement guanine (cytosine in sense chain); and "A" indicates an extended reverse complement thymine (adenine in the sense chain). This shows the mass spectrum of the post-ejection SFP.
(TIF)

**S1 Table. Percent contents of GC and AT bases in the ITS1-5.8S-ITS2 sequences of multiple genotypes of *O. sinensis*.**
(DOCX)

**S2 Table. GenBank accession numbers that are hyperlinked to GenBank database, *H. sinensis* strain, *O. sinensis* isolate, amplicon clone information, and sequence ranges for the 52 ITS sequences analyzed phylogenetically in Fig 2.**
(DOCX)

**S3 Table. Mass intensity ratios of the SNP peaks of transition and transversion mutation genotypes in the SFPs prior to and after ascospore ejection and of developmental failure.** Note: Peak G represents GC-biased Genotype #1 *H. sinensis*; Peak A indicates AT-biased

genotypes of *O. sinensis* (Fig 1). Peaks C and T denote 2 transversion mutation genotypes of unknown upstream and downstream sequences. "↑↑↑↑" denotes significant increases in intensity ratios greater than four-fold compared to that in the pre-ejection SFP. "−" means that one of the allelic peaks was missing and no ratio could be calculated.
(DOCX)

**S4 Table. Mass intensity ratios of the SNP allelic peaks of transition and transversion mutant genotypes in SFPs prior to and after ascospore ejection and of developmental failure.** Note: Peak T represents AT-biased Genotypes #5–6 and #15 of *O. sinensis*, and Peak C indicates AT-biased Genotype #4 of *O. sinensis* (*cf.* Fig 1). Peak G was nearly flat and negligible. Peak A denotes a transversion mutation genotype of unknown upstream and downstream sequences. "↔" indicates no significant change (within 20% variation), and "↑" denotes increases in intensity ratios less than two-fold compared to that for the pre-ejection SFP. "−" means that one of the allelic peaks was missing, and no ratio could be calculated.
(DOCX)

**S5 Table. Mass intensity ratios of the SNP allelic peaks of transition and transversion mutant genotypes in fully and semi-ejected ascospores.** Note: Peak G represents GC-biased Genotype #1 *H. sinensis*; Peak A indicates AT-biased genotypes of *O. sinensis* (*cf.* Fig 1). Peaks C and T denote 2 transversion mutation genotypes of unknown upstream and downstream sequences. "↔" indicates no significant change (within 20% variation) compared to the intensity ratios for fully ejected ascospores. "−" means that one of the allelic peaks was missing, and no ratio could be calculated.
(DOCX)

**S6 Table. Mass intensity ratios for the SNP allelic peaks for multiple transition and transversion mutation genotypes in the fully and semi-ejected ascospores.** Note: Peak C represents AT-biased Genotype #5 of *O. sinensis*, and Peak T indicates other AT-biased genotypes of *O. sinensis* (*cf.* Fig 1). Peaks G and A denote 2 transversion mutation genotypes of unknown upstream and downstream sequences. "↑" indicates a significant increase in the intensity ratio less than two-fold compared to that of the fully ejected ascospores. "−" means missing one of the allele peaks, and no ratio could be calculated.
(DOCX)

**S5 Fig. Reproduction of Fig 1 of [31] as requested by Reviewer #1: Cultivation of mature *Cordyceps sinensis* specimens in paper cups and collection of ascospores.** Mature *Cordyceps sinensis* specimens were cultivated in our Xining laboratory (altitude of 2,254 m) (SR1(A) Fig). The fully ejected ascospores were collected using double layers of autoclaved weighing papers (SR1(B) Fig). Numerous semi-ejected ascospores adhering to the outer surface of an ascus (SR1(C) Fig) during the massive ejection of ascospores. The stromal fertile portion (SFP) densely covered with numerous ascocarps is labeled with "]".
(ZIP)

## Acknowledgments

The authors are grateful to Prof. Mu Zang, Prof. Zong-Qi Liang, Prof. Ru-Qin Dai, Prof. Ying-Lan Guo, Prof. Ping Zhu, Prof. Zhao-Lan Li, Prof. Yu-Guo Zheng, Dr. Jia-Gang Zhao and Dr. Yan-Jiao Zhou for consultations, and Prof. Xin Liu, Prof. Hai-Feng Xu, Ms. Shao-Li Ma, Ms. Ming Yang, Mr. Wei Chen, Mr. Tao-Ye Zheng, Mr. Jin-Jin Li, and Mr. Yu-Chun Zhou for their assistance.

## Author Contributions

**Conceptualization:** Jia-Shi Zhu.

**Data curation:** Ling Gao, Yi-Sang Yao.

**Funding acquisition:** Yu-Ling Li.

**Investigation:** Yi-Sang Yao, Xiu-Zhang Li, Zi-Mei Wu, Ning-Zhi Tan, Zhou-Qing Luo, Jian-Yong Wu.

**Methodology:** Yu-Ling Li, Yi-Sang Yao, Xiu-Zhang Li, Jia-Shi Zhu.

**Project administration:** Wei-Dong Xie, Jia-Shi Zhu.

**Resources:** Yu-Ling Li, Zi-Mei Wu, Ning-Zhi Tan, Jian-Yong Wu.

**Supervision:** Wei-Dong Xie, Jia-Shi Zhu.

**Validation:** Yi-Sang Yao.

**Visualization:** Ling Gao, Zi-Mei Wu, Ning-Zhi Tan, Zhou-Qing Luo.

**Writing – original draft:** Jia-Shi Zhu.

**Writing – review & editing:** Yu-Ling Li, Jia-Shi Zhu.

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
