## [Decision Letter · Decision Letter 0]

30 Mar 2023

PONE-D-22-28022Altered GC- and AT-biased genotypes of Ophiocordyceps sinensis in the stromal fertile portions and ascospores of natural Cordyceps sinensisPLOS ONE

Dear Dr. Jia- Shi- Zhu,

Thank you for submitting your manuscript to PLOS ONE. After careful consideration, we feel that it has merit but does not fully meet PLOS ONE’s publication criteria as it currently stands. Therefore, we invite you to submit a revised version of the manuscript that addresses the points raised during the review process, as you will read them below this letter.

Please make sure (when responding to the reviewer comments) to  amend the following key issues; your abstract, language, reference style, objective of your study, results (including reducing tables from text), proper discussion of results and conclusion, without forgetting strict adherence to PLOS ONE format.==============================

We look forward to receiving your revised manuscript.

Kind regards,

Elingarami Sauli, PhD

Academic Editor

PLOS ONE

“This research was supported by a grant from the Science and Technology Department of Qinghai Province, China, grant number 2021-SF-A4 “Study on key technologies of conservation of natural resource and industrial upgrading of Cordyceps sinensis”, the major science and technology projects in Qinghai Province.”

Reviewers' comments:

Reviewer's Responses to Questions

**Comments to the Author**

1. Is the manuscript technically sound, and do the data support the conclusions?

Reviewer #1: Yes

Reviewer #2: Yes

2. Has the statistical analysis been performed appropriately and rigorously? 

Reviewer #1: Yes

Reviewer #2: Yes

3. Have the authors made all data underlying the findings in their manuscript fully available?

Reviewer #1: Yes

Reviewer #2: Yes

4. Is the manuscript presented in an intelligible fashion and written in standard English?

Reviewer #1: Yes

Reviewer #2: No

5. Review Comments to the Author

Reviewer #1: The authors described genotypes of Ophiocordyceps sinensis - insect-fungal complex. The work is novel. It is based on solid experimental background. Though it is hard to read due to redundancy in technical details, and not clear overall presentation. As I understand, the text was revised and reorganized after previous submission. It has an accompanied paper that complements the O.sinensis study.

However, the manuscript is still too large. Figures from 5 to 9 are very similar. May move them to a Supplement as well as other redundant text.

The presentation is not clear for reader with common biological educational background. It missed important details on the object of the study.

The reference style is not appropriate – bulk citations everywhere! It should be not more than 2-3 references together in the text. Otherwise need a phrase, add details, cite the references separately.

See - [6‒10], [ 7‒25] and multiple other similar in-text citations ([6‒16,19‒21,26‒38] ).

The Abstract is unclear. It should have some description of the object (it is given only later in the main text).

The ‘Objective’ of the Abstract should be extended to state the problem – it is rather about insect-fungal interactions and association with genotypes. The problem seems be to narrow (only about “SFP densely covered with numerous ascocarps and ascospores”?.

Line 44: “cultivated in our laboratory (altitude 2,200)” – need details about laboratory, is it unique place or could be cultivated anywhere at same altitude? What is the laboratory?

Line 54: “mutation alleles of unknown sequences” – unclear, rephrase

Line 64: the keywords could be more precise, exhaustive, and not so long. Comment on MassARRAY (it is not in the text).

Citations are in bulk. Please cite separately, add details, new sentences.

Even in the phrase (line 74) “Modern pharmacological studies … [2-5]” need name the studies, what is modern and recent?

Line 77: “≠” – this sign is from mathematics. Explain by words about the terminology, that it differs, or not equal.

General short names “C. sinensis” might be mixed with tea plant Latin name. It is worthy to repeat all the designation in this area again, not use new abbreviations.

Line 89: “[ 7‒25]’ such citation is not appropriate. Give necessary details, add a phrase. Or remove redundant references if not used further in the text.

Line 99: “widely accepted because the evidence meets the first and second criteria of Koch’s postulates” – please name the postulate, give a reference. Phrase “widely accepted” should be confirmed.

Line 104: “Many studies…” – name these studies. The references at the end of the sentence again are redundant ([6‒10,13,15‒16,37,47‒54]).

Line 111: “Genotypes #1 [28,30,32,39], #3 [55‒56], #4…” - it is not clear what this terminology means (#1, #2..) – is it commonly accepted, or designed by the authors?

Some new figure with map of geographic origins would be helpful. And some description of geographic and environment conditions.

Line 117: “[61 (the companion paper)]” – such citation is not appropriate. May refer to this paper by other way, not using complex parentheses.

Line 134: Section ‘Reagent; could be after section ‘Collection…’ in the text. Logically it should be after Methods description.

Line 140: MassARRAY – should be commented about the tool, complex abbreviation.

Line 144: “were purchased in local markets” – need details – condition, location of the markets, how stable such specimens. Where the specimens were collected before, transported?

Line 151: “(cf. Fig 3A below)” – the figures numbering should start from 1. 1,2, then 3. Please rearrange.

Line 155: “Fig 1 of the companion paper [61])” – again it is not clear. Refer first on the figures in the same text. May add some references to a Supplement to the paper, or other published paper. It is just not clear for reader.

Line 159: “The humidity..” – need give exact values.

Line 195: “The primers Hsprp1, Hsprp3, ITS4, P1, P2 and P4…” – these nomenclature is unclear. Need comment first, provide details about known genes and primers.

Figure 1 should be as an additional figure. First should be information about the object, known genotypes. Might be some photos as in Fig.3

Now Figure 1 gives not information about AT and GC content.

Line 207: “Genotypes #1‒3 and #13‒14 are GC-biased..” – it is not visible.

Line 215: “GenBank” – need exact link to access the data in Genbank

Details of phylogenetic analysis are too common (and too many just bioinformatics references). It is better to describe known strains.

Line 252: “iLEX Gold cocktail” – need add details.

Figure 3 is good, but has too many panels. Please make it more compact to fit to one page. Separate Figure 3 onto smaller figures.

Figures 4,5,6,7,8,9 are very similar. May keep one-two figures in the main text, and more other figures to a Supplement (new Supplementary material file, or to online resource)

Discussion section is rather technical, and it has again redundant references.

The section (starts from line 807) “True causative fungus/fungi of natural C. sinensis” is more interesting. That part of text could be in the beginning of paper.

Conclusion section is rather sort.

The dynamic alterations of genotypes is hypothesized, but not proved in current study.

Could keep it as the hypothesis. It needs lager statistics and a greater number of specimens to prove anyway.

The reference list is too long, as well as overall paper size. It could be reduced without loss of information but make presentation more clear and convenient. I suggest reformatting the text, remove up to 20% of reference, remove repeated figures with MS plots (or place it to a Supplementary file)

Reviewer #2: I have read your paper, some suggestion are as below:

I think the biggest problem with this paper is breaking up a system work into multiple paper for publication. I don't think this is a convincing reason that the amount of work is too much to publish as a paper. the author should give a persuasive reply.

Other suggestions are as follows:

1. The English in the paper need to be modified by English experts.

2. Line 143-173: The sample collection section is too long and tedious, you need to be simplified.

3. Line 144-145：Elevation information needs to be added to two sampling point.

4. Line 154: The latitude and longitude information needs to be added to your laboratory.

5. Line 214-223：The method and result are confused, please separate them.

6. The combination of small pictures in Figure 3 does not match，Please arrange the small diagram in Figure 3 properly. Additionally, some of pictures don't have scale label.

7. Line 267: Where is the ascocarp? please mark it in the Figure 3.

8. Why are some of small pictures in Figure 3 in color and some in black and white?

9. I think Figure 4A, 4B, 4C and 4D can be combined to one picture, the integrated picture is similar to Figure 4D. And also, Figure 4 and Figure 5 are merged into one figure. The same to Figure 6 and Figure 7, Figure 8 and Figure 9.

10. The discussion section can be can deleted partly or simplified, otherwise it is too long and tedious.

11. line 641-648: The layout of the discussion section is different from the rest of the section, and I recommend deleting lines 641 to 648.

12.Please check all references for consistency.

6. PLOS authors have the option to publish the peer review history of their article (what does this mean?). If published, this will include your full peer review and any attached files.

Reviewer #1: No

Reviewer #2: No

---

## [Author Response · Author response to Decision Letter 0]

22 May 2023

We have included separate files and answered the reviewers’ comments point-to-point. Please see the files of (1) 2023-5-8 Responses to Reviewer#1 and (2) 2023-5-8 Responses to Reviewer#2.

---

## [Editor Report · Decision Letter 1]

25 May 2023

Altered GC- and AT-biased genotypes of Ophiocordyceps sinensis in the stromal fertile portions and ascospores of natural Cordyceps sinensis

PONE-D-22-28022R1

Dear Dr. Jia-Shi Zhu,

We’re pleased to inform you that your manuscript has been judged scientifically suitable for publication and will be formally accepted for publication once it meets all outstanding technical requirements.

Kind regards,

Elingarami Sauli, PhD

Academic Editor

PLOS ONE
---

## [Editor Report · Acceptance letter]

31 May 2023

PONE-D-22-28022R1 

Altered GC- and AT-biased genotypes of *Ophiocordyceps sinensis* in the stromal fertile portions and ascospores of natural *Cordyceps sinensis*

Dear Dr. Zhu:

I'm pleased to inform you that your manuscript has been deemed suitable for publication in PLOS ONE. Congratulations! Your manuscript is now with our production department. 

Kind regards, 

on behalf of

Dr. Elingarami Sauli 

Academic Editor

PLOS ONE